# The economically optimal warming limit of the planet

F. Ueckerdt[1], K. Frieler[1*], S. Lange[1], L. Wenz[1,3,5], G. Luderer[1], A. Levermann[1,2,4]

[1]Potsdam Institute for Climate Impact Research, Potsdam, Germany.

[2]Columbia University, New York, USA.

[3]Mercator Research Institute on Global Commons and Climate Change, Berlin, Germany.

[4]Institute of Physics, Potsdam University, Potsdam, Germany.

[5]Department of Agricultural and Resource Economics, University of California, Berkeley, USA.

*Correspondence to: katja.frieler@pik-potsdam.de

## Abstract

Both climate-change damages and climate-change mitigation will incur economic costs. While the risk of severe damages increases with the level of global warming (Allen et al., 2018; Dell et al., 2014; IPCC, 2014b; Lenton et al., 2008), mitigating costs increase steeply with more stringent warming limits (IPCC, 2014a; Luderer et al., 2013a; Rogelj et al., 2015). Here we show that the global warming limit that minimizes this century's total economic costs of climate change lies between 1.9 and 2°C, if temperature changes continue to impact national economic growth rates as observed in the past and if instantaneous growth effects are neither compensated nor amplified by additional growth effects in following years. The result is robust across a wide range of normative assumptions on the valuation of future welfare and inequality aversion. We combine estimates of climate change impacts on economic growth for 186 countries (applying an empirical damage function from Burke et al., 2015) with mitigation costs derived from a state-of-the-art energy-economy-climate model with a wide range of highly-resolved mitigation options (Kriegler et al., 2017; Luderer et al., 2013b, 2015). Our purely economic assessment, even though it omits non-market damages, provides support for the international Paris Agreement on climate change. The *political* goal of limiting global warming to "well below 2 degrees" is thus also an *economically* optimal goal given above assumptions on adaptation and damage persistence.

## 1. Introduction

"Holding the increase in the global average temperature to well below 2°C above preindustrial levels and pursuing efforts to limit the temperature increase to 1.5°C" is a central element of the global climate agreement reached in Paris in December 2015 (UNFCCC, 2015). This political goal builds on the scientific insight that a global warming beyond 1.5–2°C poses risks of potentially severe impacts such as insecure food and drinking water supply (Allen et al., 2018; IPCC, 2014b), threatened biodiversity (Dawson et al., 2011; Willis and Bhagwat, 2009), large-scale singular events (Lenton et al., 2008; Schellnhuber et al., 2016), displacement (Hsiang and Sobel, 2016), health impacts (Burke et al., 2018; Carleton, 2017; Matthews et al., 2017) or human conflict (Hsiang et al., 2013a; Schleussner et al., 2016). Many of these risks and their societal consequences are difficult or even impossible to capture in economic terms or market costs.

Here we focus on the direct impacts of global warming (damages and mitigation) on economic output (GDP). Taking a purely economic perspective that omits non-market damages, we derive the optimal warming limit of the planet by minimizing this century's (2015–2100) market costs of climate change. The analysis combines mitigation cost estimates from a detailed integrated assessment model, also referred to as energy-economy-climate model, with a country-specific empirically-based damage estimation. Following Burke et al., 2015, we assume that the observed relation of economic damages and annual temperatures of a country remains valid for the future and that damages are one-year growth reductions (neither recovery nor additional growth impact in following years).

Cost-benefit integrated assessment models (IAM) such as FUND, PAGE and DICE (Anthoff and Tol, 2014; Hope, 2013; Nordhaus, 2014, 2010) typically combine a stylized representation of mitigation strategies with "damage functions", which aggregate the economic costs from climate impacts as a function of the global warming. Past representations of climate change impacts have been found to suffer from several limitations (Pindyck, 2013; Revesz et al., 2014). In particular, it has been criticized that damage functions used by the major IAMs and for the computation of the Social Costs of Carbon (SCC) do not reflect recent empirical estimates of climate induced damages.

Recent work contributes to closing this gap. Moore et al., 2017 update the FUND model damage function by incorporating the most recent empirical estimates for agriculture and find more than a doubling of SCC. Ricke et al., 2018 estimate country level SCC using empirical damage estimates (Burke et al., 2015; Dell et al., 2012). Moore and Diaz, 2015 implement empirical estimates of temperature effects on GDP growth rates in the DICE model. Howard and Sterner, 2017 conduct a meta-analysis of global climate damage estimates and use a synthesized temperature-damage relationship to replace the DICE-2013R damage function. Drouet et al., 2015 derive optimal carbon budgets from combining a range of energy-economy-climate model results with damage functions that are based on climate-impact estimates.

Here we complement existing estimates by combining country-specific empirical damage estimates with mitigation cost estimates from a more detailed integrated assessment based on the energy-economy-climate model REMIND (Kriegler et al., 2017; Luderer et al., 2013b, 2015). We estimate climate damages from annual gridded temperature data (0.5° x 0.5° resolution) for 186 countries based on the empirical relation between temperature deviations and economic growth rates derived in Burke et al., 2015. The REMIND model couples an economic growth model with a bottom-up, technology-explicit energy system model and a simple climate model. Mitigation cost estimates are thus based on a modelling system with detailed and explicit process detail (e.g. ~50 energy conversion technologies, as well as a sectoral representation of non-CO2 greenhouse gas emissions), in contrast to exogenous abatement cost functions in cost-

benefit integrated assessment models depicting mitigation strategies in a more stylized manner. In their pioneering work, Burke et al., 2015 derive an empirical relation of annual historical temperature deviations and GDP changes based on country-specific data for 50 years (1960-2010) and 166 countries (which we then apply for 186 countries). Their regression analysis captures the aggregated climate-related impacts across all economic sectors that contribute to a country's GDP changes. Burke et al., 2015 find that growth rates change concavely in temperature, i.e. cold-country productivity increases as annual temperature increases, whereas warm-country productivity decreases and this decline accelerates at higher temperatures (see Fig. A4). Damage aggregates across countries show that losses exceed benefits such that global damage estimates are high (>20% of global GDP in 2100 under RCP8.5, see Fig 1a). For more details on the calculation of damage costs see appendix A2.

Burke et al., 2015 reconcile micro and macro-level observations by accounting for non-linearities at the macro-scale (Sterner, 2015). There are many empirical impact studies on the micro-level (for e.g. agriculture (Moore et al., 2017; Schlenker and Roberts, 2009), electricity (Wenz et al., 2017), labor productivity (Zhang et al., 2018; Zivin and Neidell, 2010), which find high and often strongly non-linear economic damages from climate change. Burke et al., 2015 demonstrate how disruptive changes on the micro-level can translate into a smooth non-linear GDP-temperature effect on the macro-level.

The macro-level estimates by Burke et al., 2015  allow for deriving aggregated economic estimates of both temperature-induced losses and benefits across economic sectors and potential impact channels, e.g. impacts on health costs, labor productivity, or crop yields, without relying on an explicit representation of the underlying processes or sector-specific micro data. The resulting relation is robust for subsets of countries (poor and rich countries; agricultural producing and less agricultural producing countries; also see Fig. A4).

The statistical evidence presented by Burke et al., 2015 challenges standard economic modelling and has initiated a highly relevant debate about alternative approaches (Letta and Tol, 2018; Moore and Diaz, 2015) and potential methodological refinements (Burke et al., 2016; Carleton and Hsiang, 2016; Hsiang et al., 2017; Kalkuhl and Wenz, 2018; Mendelsohn, 2017; Ricke et al., 2018). Further research will include more sector-specific information and process-based understanding to refine the empirical analysis by disentangling different economic impact channels. Even though Burke et al., 2015  do not provide the final word on impacts of temperature changes, their approach creates a novel opportunity for a necessary next step towards a comprehensive assessment of the costs and benefits of climate action that we undertake in this analysis. The empirically estimated temperature-GDP relation now allows to carry out a comparison of the costs that will arise from climate change impacts and costs to avoid future climate change on the basis of i) empirically-based damage estimation combined with ii) mitigation cost estimates from a detailed energy-economy-climate model.

The empirically derived relationship is in principle comprehensive regarding all processes contributing to the GDP-temperature linkage and even implicitly covers market side effects of mainly non-market damages such as ecosystem degradation or changes in water quality and food supply. The approach does however not allow for explicitly resolving these processes and may thus neglect potential future changes in their relevance. We neither account for potential future adaptation mechanisms that might dampen the observed sensitivity nor for possible amplifications, for example, due to a potential destabilization of societies (Hsiang et al., 2013a). The assumption of robustness of the relationships is supported by i) its stability across the historical period where past warming did not induce notable adaptation to the considered

economic impacts (Burke et al., 2015) and ii) its stability across the wide range of countries with very different climatic and socio-economic conditions. In addition, the assumption is more reliable under low levels of global mean temperature change, which turn out to be the most relevant for our study (see Appendix A3).

Apart from normative parameters such as the pure rate of time preference and inequality aversion, the estimates of optimal warming limits and SCC depend on evidence-based assumptions on the persistence of climate damages. Following Burke et al., 2015, we assume a "one-year-growth effect", i.e. an instantaneous decrease of an annual growth rate due to a temperature change in year $t$, which is neither compensated nor increased by additional growth effects in the following years $t + i$. Small changes in annual growth rates in one year can accumulate to high damage levels over time and thus climate impacts will have long-lasting impacts on GDP (Fankhauser and S.J. Tol, 2005; Moore and Diaz, 2015; Moyer et al., 2014). Burke et al., 2015 attempt try to analyse the persistence of empirical damages by allowing for lagged responses, but the observational data do not provide enough statistical power to robustly estimate neither magnitude nor sign of the additional parameters introduced into the regression (Extended data figure 2c in Burke et al., 2015). In general, the question of persistence and associated magnitude of climate damages remains both very relevant and open (see a more detailed discussion in appendix A6). The ways in which climate change interacts with economic productivity, capital or labor stocks are complex (Huber et al., 2014). Existing literature points to both level and growth effects (Burke and Tanutama, 2019; Dell et al., 2012; Felbermayr and Gröschl, 2014; Hsiang and Jina, 2014; Hsiang, 2010; Newell et al., 2018; Piontek et al., 2018) and it requires more empirical studies, meta analyses and process understanding to draw robust conclusions. If level effects prevail, our assumption of one-year-growth effects overestimates damages and distorts our analysis towards too low warming limits. If by contrast persistent growth effects prevail, one-year-growth effects would underestimate damages and required mitigation action. The scope of this paper is to demonstrate the implications of combining empirical-based one-year-growth effects with climate mitigation costs to infer about optimal global warming levels.

## 2. Materials and Methods

We combine the damage estimates with climate change mitigation costs from the REMIND energy-economy-climate model, which provides an integrated and explicit representation of the macro-economies and energy systems of 11 world regions until 2100. REMIND captures a particularly wide range of climate change mitigation options as well as relevant path dependencies with substantial process detail, allowing for a quantification of mitigation costs for warming limits down to even below 1.5°C by 2100 (Luderer et al., 2013a; Rogelj et al., 2015). More information on REMIND can be found in the appendix A4 and in a model description paper and wiki (Luderer et al., 2015, *https://www.iamcdocumentation.eu/index.php/IAMC_wiki).* Estimating both climate change damages and mitigation costs is subject to uncertainties and normative assumptions (Drouet et al., 2015; Kopp and Mignone, 2012; Revesz et al., 2014). Here we account for i) uncertainties in the climate system's response to emissions by using simulations from twelve General Circulation Models (GCMs) generated within the Coupled Model Intercomparison Project Phase 5 (CMIP5, Taylor et al., 2012) and ii) uncertainties in the GDP response to temperature changes by accounting for the statistical uncertainties of the regression parameter inBurke et al., 2015. In addition, we broadly vary the assumptions on the

normative weighting of future costs (pure rate of time preference) and inequality (inequality aversion).

Uncertainty in results of energy-economy-climate model is typically analysed by means of multi-model ensembles and in dedicated model-intercomparison projects (Krey et al., 2014; Kriegler et al., 2014; Luderer et al., 2016), largely reflecting the dominant importance of structural differences across models. There are about one dozen well-established models in the global community of detailed integrated-assessment models, the contributions of which are a crucial foundation of the transformation pathway chapters as well as the summary for policy makers sections of the IPCC reports (Allen et al., 2018; IPCC, 2014a). Accounting for structural model uncertainty on the mitigation cost side in the context of this analysis would require running harmonized scenario sets with multiple models for a broad range of warming limits going down to 1.5°C. While this is beyond the scope of this analysis, it has been shown that mitigation costs from REMIND are close to the median of cost estimates from other models that have contributed to the IPCC AR5 scenario ensemble (Clarke et al., 2014). Kriegler et al., 2015 introduced diagnostic indicators based on a multi-model analysis with stylized scenarios to characterize model behavior. All cost indicators show significant differences across models, while the REMIND model results are close to the across-model median. This can also be seen in appendix figure A6, which compares REMIND mitigation costs for different warming levels with recent scenario results of the IPCC SR15 (Allen et al., 2018). Hence, the REMIND model can be regarded as a representative IAM in a single-model approach. We discuss the implications of mitigation cost uncertainties when presenting the results and in appendix A5.

We combine two partial analyses, for mitigation and damage costs. Not integrating them on the system level neglects three main interactions. First, climate induced reductions of economic productivity and associated reductions in energy demand would lead to reduced emissions without explicit mitigation measures (Bastien-Olvera, 2019; Woodard et al., 2019), while climate impacts might reduce financial resources for climate mitigation. Second, climate impacts might change the future energy supply by their impact on renewable potentials and temporal variability (hydro, biomass, solar or wind power) and extreme weather events on energy infrastructure such as storm-induced transmission breakdowns and power outages or limited cooling water for nuclear or thermal plants (for further literature see this review: Cronin et al., 2018). Third, we did not analyse to what extent a full internalization of climate damages would shift the welfare optimal timing of mitigation to avoid short-term damages compared to a mitigation scenario that focuses only on limiting global warming. Reflecting those various interactions in an integrated study is complex and a future task to the scientific modeling communities. Accounting for these interactions requires a better process-understanding by which channels climate impact unfold and more empiric quantifications following pioneering work for individual processes e.g. energy demand (Bastien-Olvera, 2019; Woodard et al., 2019). Currently, the macro-level temperature response identified by Burke et al., 2015 could not be broken down to individual processes. It even seems difficult and premature to conclude on the overarching magnitude or sign of climate impacts on the energy transitions and mitigation costs.

For deriving damage costs, we estimate climate-induced annual GDP losses for 186 countries based on annual country-specific temperature projections from twelve GCMs, three different climate change scenarios (Representative concentration pathways: RCPs), and one no-further-warming scenario (Appendix A1.2). For the reference economic and demographic developments (country-specific GDP and population without climate change) we adopt the "middle-of-the-

road" shared-socio-economic pathway (SSP2, O'Neill et al., 2015), and use the four other SSPs as sensitivity cases. The temperatures are population-weighted based on spatially highly-resolved (0.5° x 0.5°) dynamic population projections (Jones and O'Neill, 2016a) (Appendix A1.3) to ensure that the analysis is not distorted by extreme temperatures in deserted areas. When calculating the temperature impact on annual country-specific growth rates we distinguish between rich and poor countries by choosing the respective empirical regression parameters from the "base" case in lBurke et al., 2015 (see also Appendix equations S7-S9). The extrapolation of the observed temperature-growth relation yields globally aggregated annual climate-induced GDP losses that amount to up to 40% in 2100 under the highest emissions scenario RCP8.5 compared to SSP-specific baseline scenarios of economic development (Fig. 1a, shown for 4 selected GCMs that represent the range within the ensemble of 12 GCMs, and based on the median specification of regression parameters of the empirical analysis (Burke et al., 2015)). These losses are reduced to ~10% under the strong mitigation scenario RCP2.6.

Globally aggregated mitigation costs (relative GDP losses compared to a no-climate-change reference scenario) were derived for ten different scenarios with maximum warming limits of 1.6°C to 4.2°C (Fig. 1b) from optimal transition pathways of the global economy and energy system calculated by the REMIND model. Note that the corresponding end-of-century warming levels (in 2100) go down to well below 1.5°C. The underlying mitigation scenarios assume global cooperative action with harmonized greenhouse gas emissions pricing as of 2020 and a broad portfolio of low-carbon technologies, including carbon capture and storage (CCS) also in combination with bioenergy (BECCS), thereby generating negative emissions. We assume that, in line with the principle of "common but differentiated responsibilities and respective capabilities"(UNFCCC, 1992), a financial transfer scheme is in place that distributes mitigation costs among all countries in proportion to their annual GDP, while maintaining a cost-minimizing distribution of physical emission reduction efforts across regions. Mitigation costs (globally aggregated for 2015–2100) increase steeply with warming limits decreasing towards 1.5°C (Fig. 1c).

Total costs of climate change in a particular scenario are estimated as the associated social welfare loss relative to a scenario without climate change (in our case the SSP2 baseline scenario).
The social welfare function $W$ aggregates annual country-specific per-capita utility $U(t, i)$ for all years $t \in [2015, 2100]$ and all countries $i \in [1, 186]$ with respective populations $n_i(t)$:

$$W = \sum_{t=2015}^{2100} \sum_{i=1}^{186} n_i(t)U(t,i)(1 + \delta)^{-t}$$

where $U(t, i)$ is an isoelastic utility function of per-capita consumption $C(t, i)$:

$$U(t,i) = \begin{cases} \dfrac{C(t,i)^{1-\varepsilon}}{1-\varepsilon}, & \varepsilon \neq 1 \\ \ln(C(t,i)), & \varepsilon = 1 \end{cases}$$

The two normative parameters pure rate of time preference ($\delta \in [0, 0.04]$) and inequality aversion ($\varepsilon \in [0, 2.5]$) determine how consumption losses are weighted in time and across countries when aggregating global welfare. Increasing the pure rate of time preference $\delta$ in equation (1) gives higher weights to present compared to future generations' utilities and hereby shifts the optimal warming towards higher values (Fig. 1d) because a major share of

mitigation costs incurs already in the next years while the bulk of damage costs occurs in the second half of the century (Fig. 1a and b).

At the same time, climate change impacts vary across countries at different levels of economic development. With increasing inequality aversion $\varepsilon$ the consumption of a poorer individual is weighted more strongly than the consumption of richer individuals, i.e. utility as a function of consumption (equation (2)) becomes more concave. For $\varepsilon=0$ the utility function is linear and thus does not account for inequality in wealth levels (inequality neutrality). For $\varepsilon=1$ the utility function is logarithmic and thus relative changes in consumption receive equal weight, i.e. doubling consumption creates the same welfare gain for rich and poor individuals. Inequality aversion works both across countries and in time, as it also affects the weighting of future, potentially richer generations relative to present ones. Spatial and temporal inequalities push the optimal warming towards opposite directions. Climate impacts tend to be higher in poor countries, hereby increase inequality and thus call for higher mitigation ambition to decrease optimal warming. Conversely, future generations will be richer and thus allowing for higher future impacts by reducing the current generation's mitigation burden decreases inequality and increases optimal warming.

We approximate country-specific mean per-capita consumption in terms of per-capita GDP values, which corresponds to the assumption of an invariant savings rate. The separately estimated GDP losses from detailed analyses of both climate impacts and climate mitigation are combined by reducing the reference GDP (without climate change) successively by the two relative GDP losses. Before we can combine them, GDP losses from damages and mitigation need to be harmonized. Relative GDP losses from damages are estimated for the "middle-of-the-road" scenario SSP2 as region-specific GDP and population developments in this scenario are similar to those in the REMIND no-climate-change reference scenario that is used for estimating mitigation costs. RCP6.0 is excluded from the derivation of the optimal warming limits as its emission trajectory is qualitatively different from the other RCPs and regarded less representative for the range of scenarios considered within the IPCC Fifth Assessment Report. Relative GDP losses from damages for the remaining RCPs are interpolated to the ten global warming limits of the mitigation cost scenarios such that mitigation and damage data refer to a consistent set of global warming limits (Appendix A2). Finally, the climate-induced losses in social welfare for 10 different global warming limits are interpolated with cubic splines (see lines in Fig. 1c). The minimum of the interpolating function marks the optimal warming (depending on the GCM and normative choices of pure rates of time preference and inequality aversion).

## 3. Results

Optimal global warming limits (Fig. 2a: GCM median values, median damage parameter specification from Burke et al. (Burke et al., 2015), SSP2 scenario) are below 2°C across a wide range of parameters, in particular for values typically used in the economic literature (see shaded area). The IPCC-AR5 identified *"a broad consensus for a zero or near-zero pure rate of time preference"* (IPCC, 2014a), which we interpret as <1% p.a. values. This is also in line with a recent expert survey giving a median value of 0.5% (Drupp et al., 2018). Inequality aversion values $\varepsilon$ typically range between 0.5 and 2.5 (Anthoff et al., 2009; IPCC, 2014a; Pearce, 2003).

The median estimates of the optimal warming are robust against the choice of the normative parameters up to a pure rate of time preference of $\delta=2.5\%$ p.a. (see Fig. 2b-e). This robustness

indicates a distinct minimum in total costs of climate change at 1.9–2°C surrounded by a sharp increase of mitigation costs below 1.9°C and the ever increasing damage costs above 2°C. For more extreme combinations of high pure rates of time preference ($\delta \in [2.5, 4]$ % $p.a.$) and low inequality aversion ($\varepsilon \in [0, 0.5]$), the optimal warming rises up to ~2.6°C. In these cases, climate damages inflicted on future generations and poor countries have less weight, thus disincentivizing mitigation efforts.

Fig. panels 2b-e display the 50% confidence intervals from varying the damage parameter (grey) and the 50% confidence intervals due to deviations in the GCM ensemble (orange). The lower range of optimal temperatures remains close to median values (1.8–1.9°C). This limited impact of uncertainty is caused by steeply increasing mitigation costs below a warming limit of 1.9°C (Fig. 1c). By contrast, the upper range of optimal temperatures can reach up to ~3.4°C for very low inequality aversions, which is considerably warmer than the <2°C median values. This range is driven by uncertainties in the empirical quantification of the complex interaction of temperature changes and economic productivity, which are higher than the effect of deviations in the climate representations from the ensemble of 12 GCMs. The productivity-temperature functions that correspond to the 25% percentile of regression parameters in Burke et al., 2015 become flatter (Fig. A4) and thus impose less climate damages than the median specification. This translates into higher optimal temperatures in particular for low inequality aversions. For higher inequality aversions, the effect of these damage uncertainties decreases due to a more heterogeneous distribution of climate impacts in the 25% percentile specification. While the two temperatures that maximize productivity for rich and for poor countries (Fig. A4), are close for the median specification ($\Delta T_{max,median} = 1.1°C; T^{rich}_{max,median} < T^{poor}_{max,median}$), they deviate from one another for the 25% percentile ($\Delta T_{max,25} = 5.3°C; T^{rich}_{max,25} > T^{poor}_{max,25}$). The effective difference between the median and 25% percentile specification is even higher ($\Delta T_{max} = 6.4°C$), since the order of $T^{rich}_{max,X}$ and $T^{poor}_{max,X}$ changes, such that with increasing national temperatures rich countries benefit longer and poor countries lose much earlier in the 25% percentile specification. Climate-induced regional inequality becomes more pronounced and, if deemed unfavorable (i.e. for $\varepsilon$-values of about 2.5), the confidence interval narrows such that the upper bound of the 50% confidence interval of optimal temperatures is T<2°C for $\delta$=0% p.a. and T<2.2°C for $\delta$=1% p.a..

## 4. Discussion & Conclusion

There are limitations to the presented analysis that can pull the optimal warming estimates towards both lower and higher values. Optimal warming limits can increase if adaptation measures substantially reduce the negative effects of higher temperatures. While some studies suggest cost-efficiency of specific adaptation measures for the future (Barreca et al., 2016; Hinkel et al., 2014; Jongman et al., 2015), other studies project persistent adaptation gaps based on evidence in historic data (Burke et al., 2015; Burke and Emerick, 2016; Carleton, 2017; Carleton and Hsiang, 2016; Moore et al., 2017). The empirical analysis (Burke et al., 2015) applied in this study reports "no notable adaptation" in the observed temperature dependence of economic growth in 1960–2010.

On the other hand, while the approach aims at comprehensively assessing market costs (i.e. direct impacts on GDP), some future climate-related impacts are missed or underestimated such that optimal warming limits from a more comprehensive welfare perspective would be even

lower. Climate change also causes non-market losses and damages such as adverse effects on human mortality or biodiversity. Climate mitigation can also induce welfare-increasing co-benefits such as health impacts like improved air quality (McCollum et al., 2013; West et al., 2013; Xie et al., 2018). In addition, the representation of climate damages as a simple function of annual temperatures and GDP neglects possible supranational spillover effects and market responses (Kalkuhl and Edenhofer, n.d.; Wenz and Levermann, 2016; Willner et al., 2018). It also abstracts from interactions between less aggregated economic damages, such as losses in specific economic sectors, and bio-physical impacts, such as floods or droughts, that will only unfold with further warming (Hsiang et al., 2017). These interactions can lead to additional non-linear effects (or even natural (Lenton et al., 2008) or social tipping points such as human conflicts (Hsiang et al., 2013b; Schleussner et al., 2016)), which could increase overall damages. However, country-specific temperature fluctuations in the historic period (1960-2010) reach up to 2–3°C, which is in the same order of magnitude as future temperature changes due to climate change in the RCP 2.6 and for many GCMs also in the RCP 4.5 (see Fig.A2 and A3 and Appendix A3). We thus carefully conclude that there is only a small effect of this limitation at moderate warming levels of up to ~2.5°C (RCP 4.5), which are most relevant in our analysis. In addition, the steep increase in mitigation costs limits the lower range of optimal warming estimates and the potential effect of additional damages (as seen above for the impact of uncertainty).

Figure 1a and 1b show that the temporal distribution of damage and mitigation costs is very different. Mitigation costs reach significant cost levels early in the remaining century (2030-2040) whereas damage cost exceed mitigation costs about 20 years later (2050-2070). Discounting future costs increases the relative importance of mid-term mitigation costs and shifts results of a cost-benefit analysis towards higher warming limits. Against this background, it is surprising that the optimal warming results are robust across a relatively wide range of pure rate of time preference. The reason is that damage estimates reach very high values in the second half of the century. Burke et al., 2015, damages are higher than earlier estimates (Dell et al., 2012), because the underlying assumption of growth effects of national temperature fluctuations (as opposed to level effects) implies a long-term accumulation of damages. These damages push the cost-optimal warming limits towards the *feasibility frontier* of climate change mitigation (Luderer et al., 2013b; Rogelj et al., 2015). Feasibility frontiers mark a threshold of global warming limits at which associated mitigation costs increase steeply until further mitigation is practically infeasible (see appendix figure A6). The overall results are thus mainly determined by the high damage estimates and the location of the feasibility frontier of mitigation along the warming limits axis, which is subject to uncertainty. Most models show steeply increasing mitigation cost in between 1.5 and 2°C. A re-analysis of global mitigation cost estimates in the IPCC SR1.5 Scenario Database (Huppmann et al., 2018) (see appendix figure A6), shows (a) an uncertainty range of about +/- 0.2 °C of warming limits achievable at high cost levels, and (b) that the REMIND feasibility frontier is located in the middle of the spectrum given by the SR1.5 model ensemble, and thus can be regarded representative. This suggests that the impact of variance of detailed IAM-based mitigation costs is less than the impact of the statistical uncertainty in the damage parameters (Burke et al., 2015) that this study is based on.

Optimal warming limits can also increase if additional barriers to mitigation are included into the scope of the analysis. The mitigation scenarios in this analysis assume that emission reductions are reached cost-efficiently. The underlying transformation, of e.g. the global energy systems, requires policies such as carbon pricing schemes that cover a large share of global GHG emissions. While also an imperfect policy mix can initiate a similar transformation at comparable

mitigation costs (Bertram et al., 2015), a lack of political or societal will, partial interest groups and lobbying power, weak institutions, or insufficient international cooperation could hamper or delay a transition such that mitigation costs increase. Our analysis is meant to inform the ongoing international climate negotiations under the assumption that these barriers can be overcome.

The political and scientific debate about an adequate global warming limit is ongoing. While the Paris Agreement (UNFCCC, 2015) specifies a limit of 1.5–2°C, the "Intended Nationally Determined Contributions" of the signing countries imply a much higher warming of 2.6–3.1°C by 2100 (Rogelj et al., 2016). Building on the recent methodical advances in estimating climate change damages and mitigation costs, we show that a purely economic assessment, which assumes that temperature changes continue to impact economic productivity as observed in the past, supports the ambitious long-term temperature goal set in the Paris Agreement.

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

## Acknowledgments

**General**: Temperature observations for bias correction were accessed from the EartH2Observe server. Upscaled HYDE 3.1 population density data were supplied by Franziska Pointek. Downscaling of future national SSP based population data to the considered grid has been done by Tobias Geiger.

**Funding:** F.U. received funding from Hanley Sustainability Institute of the University of Dayton. The work was supported within the framework of the Leibniz Competition (SAW-2013-PIK-5) and has received funding from the European Union's Horizon 2020 research and innovation program under grant agreement No 641811. F.U. gratefully acknowledges START project funding from the German Federal Ministry of Education and Research (BMBF). LW gratefully acknowledges funding from the Volkswagen foundation and the Ciriacy-Wantrup fellowship porgramme of UC Berkeley.

**Author contributions:** K.F., A.L. and F.U. designed the study and coordinated the work. F.U. calculated climate change damages, combined them with mitigation cost estimates and did most of the writing. S.L. derived GCM temperatures and designed the zero-emissions and no-further-warming scenarios. G.L. provided the mitigation cost scenarios. L.W. was closely engaged in the damage calculation. All authors discussed results, commented and edited the manuscript text.

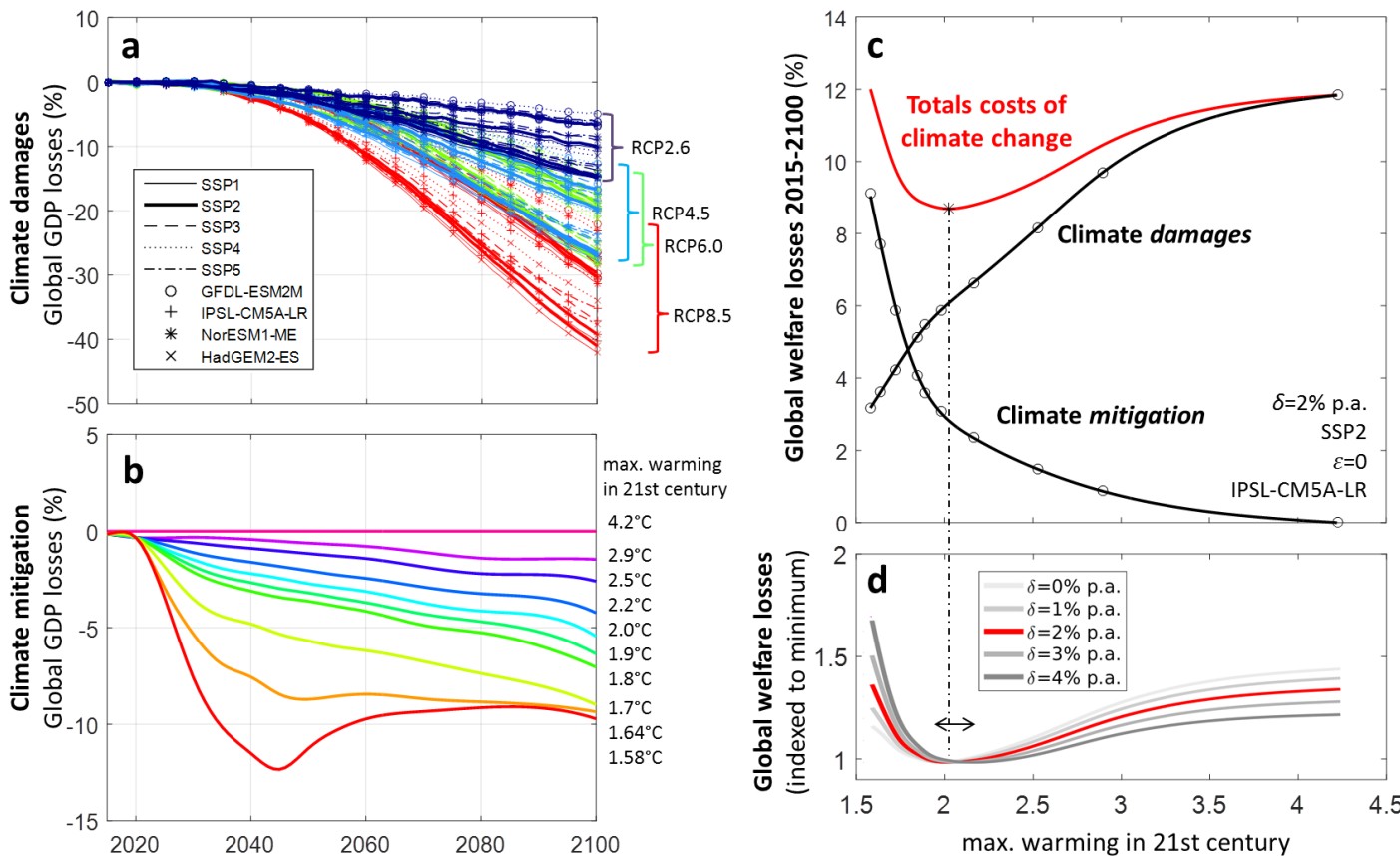

**Fig. 1**. **Deriving the economically optimal global warming limit.** (**a**) Global annual GDP losses from climate change impacts derived from the observed non-linear relationship between country-specific temperature fluctuations and GDP growth shown for 4 GCMs, 5 SSPs, 4 RCPs. Negative values correspond to losses. (**b**) Global annual GDP losses from climate change mitigation as estimated with the REMIND model(Luderer et al., 2013a; Rogelj et al., 2015) for global warming limits (color coding) from 1.6°C to 4.2°C above preindustrial levels. Negative values correspond to losses. (**c**) Cumulated global welfare losses (2015–2100) from climate damages, climate mitigation and their combined effect (total costs), as a function of global warming limits illustrated from an example scenario (SSP2, GCM: IPSL-CM5A-LR, inequality aversion $\varepsilon=0$, pure rate of time preference $\delta=2\%$ p.a.). Total costs are derived in three steps: i) Climate impacts and climate mitigation are combined by reducing the reference GDP (without climate change) successively by the two relative annual country-specific GDP losses; ii) resulting country-specific GDP pathways (with and without climate change) are translated to per-capita utility via an isoelastic utility function with varying inequality aversion; iii) resulting utilities are globally and temporally (2015–2100) aggregated to a social welfare function varying the pure rate of time preference. (**d**) Dependence of total cumulated welfare losses on pure rates of time preference. Losses are normalized by the minimum loss of each curve. Red line for $\delta = 2\%$ corresponds to red line in panel c (dashed vertical line). Cost-minimizing global warming limits slightly shift towards higher values with increasing pure rate of time preference (range indicated by arrow).

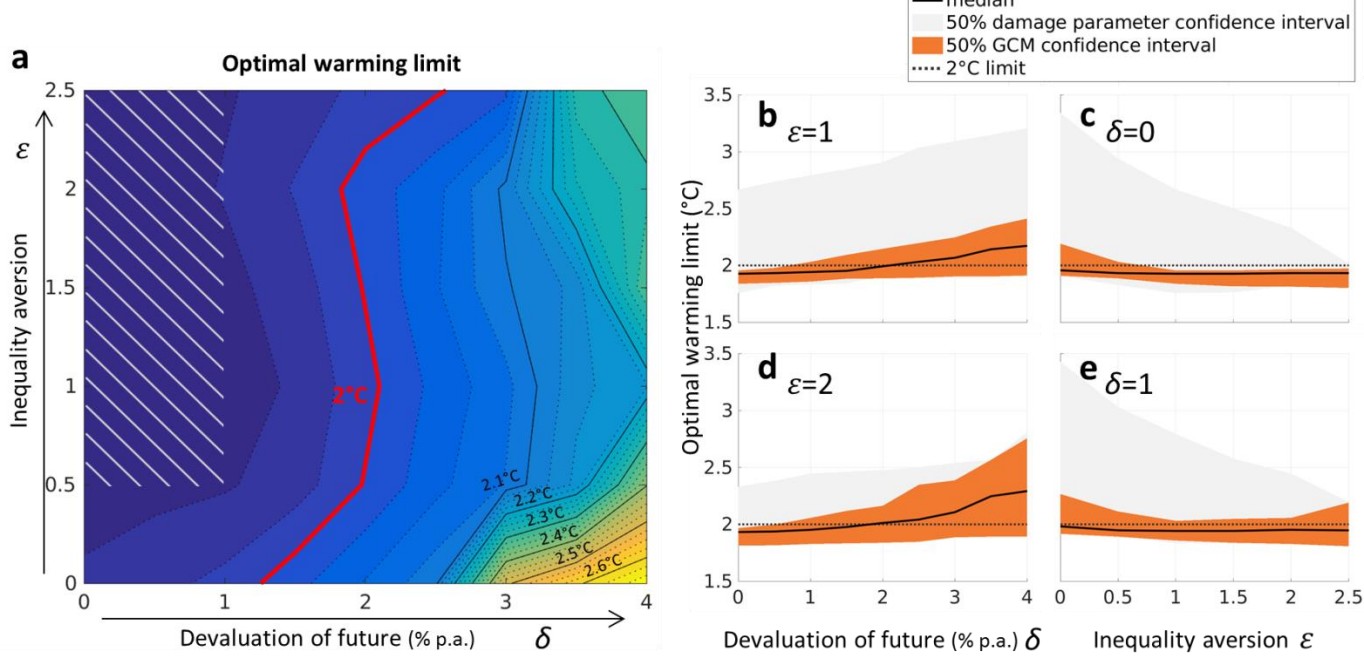

**Fig. 2. Optimal warming limits.** Maximum global mean temperature increase above preindustrial levels in the 21st century are below 2°C for a broad range of values of the normative parameters pure rates of time preference $\delta$ and inequality aversion $\varepsilon$. (a) Optimal global warming limits from the GCM median values and median damage parameter specification from Burke et al., 2015 applied for the SSP2 scenario. The shaded area marks typical literature values for both normative parameters(Anthoff et al., 2009; IPCC, 2014a). (b-e) One-dimensional cross-sections of the two-dimensional plot focused on the typical parameter range: (b) $\varepsilon$=1, (c) $\delta$=0% p.a., (d) $\varepsilon$=2 and (e) $\delta$=1% p.a., all for SSP2, indicating the median (black line), the 50% confidence interval of GCM results (grey) and damage parameter specification (orange).

**Appendix**

In Section 1 of the Appendix we report how we derive the underlying climate change data of the economic analysis, most importantly annual country-specific temperatures for different climate projections and General Circulation Models (GCMs), in Section 2 we discuss how we estimate damages from the climate data using the observed relation between country-specific temperature changes and economic growth.

### A1. Climate change data for damage calculations

Our damage calculations are based on 21st century climate change projections from phase 5 of the Climate Model Intercomparison Project (CMIP5 (Taylor et al., 2011, p.5)). More specifically, we employed monthly mean near-surface temperature data from the historical and all RCP runs done with ensemble member r1i1p1 of 12 CMIP5 GCMs (CCSM4, CSIRO-Mk3.6.0, GFDL-CM3, GISS-E2-R, MIROC5, MIROC-ESM, MRI-CGCM3, BCC-CSM1.1, GFDL-ESM2M, IPSL-CM5A-LR, NorESM1-ME, HadGEM2-ES). In this section we describe how these temperature data were bias-corrected and spatially aggregated at the country level using population density weights and how the temperatures of our no-further-warming scenario were constructed.

### A1.1. Bias correction

Annual mean near-surface temperature time series were bias-corrected with a simple delta method using observations from the Climatic Research Unit (CRU) TS3.10 dataset (Harris et al., 2013).[1] The bias correction was done on the 0.5° CRU grid, to which simulated temperature time series were interpolated with a first-order conservative remapping scheme (Jones, 1999) in order to approximately retain area mean values. Local temperature biases of ESMs were defined as deviations of local historical $1980 - 2005$ mean near-surface temperatures from the respective CRU observations, $\Delta T_i^m = \frac{1}{26}\sum_{j=1980}^{2005}(T_{ij}^{m,\text{hist}} - T_{ij}^{\text{CRU}})$, with $T_{ij}^{m,\text{hist}}$ being the mean temperature in grid cell $i$ over year $j$ simulated with ESM $m$ in the historical CMIP5 run, and $T_{ij}^{\text{CRU}}$ being the corresponding CRU TS3.10 observation. Corrected temperature space-time series $\tilde{T}_{ij}^{mp}$ for ESM $m$ and emissions pathway $p$ were then obtained by subtracting these biases from the respective raw space-time series, $\tilde{T}_{ij}^{mp} = T_{ij}^{mp} - \Delta T_i^m$.

### A1.2. No-further-warming scenario temperatures

To span a wide range of potentially optimal warming limits, our analysis requires damage estimates for climate projections well below those of RCP2.6. Temperature projections consistent with corresponding low-end emissions pathways needed to be emulated because

---

[1] In order to prevent problems arising from mismatches between the CRU land-sea mask and the country shape files used to obtain population-density weighted country mean temperatures (Section 0), we used monthly mean near-surface temperatures from the WFDEI dataset, extended to the oceans with ERA-Interim reanalysis monthly mean 2m temperatures by Emanuel Dutra for the EartH2Observe project (Weedon et al., 2011, 2014). Over land, these temperatures are equal to those of CRU TS3.10.

climate projections for such low-end emissions pathways were not done in CMIP5. This allowed us to choose a low-end emissions scenario that best suited our objectives. We decided for a *no-further-warming scenario* that follows RCP2.6 until end of 2015 and continues with no further temperature increase until 2100.

More specifically, temperature data for the no-further-warming scenario were constructed at the grid scale based on bias-corrected RCP2.6 temperatures $\tilde{T}_{ij}^{m,\text{RCP2.6}}$ from the time period of 19 years that is centered at 2015, i.e. $j \in [2006, 2024]$. These time series were linearly detrended at the grid scale such that 2006–2024 mean temperatures were preserved. Five copies of these detrended time series were then concatenated to yield 95 years' worth of temperature data covering the period 2006–2100.

### A1.3. Spatial aggregation with population density weights

Our analyses are based on annual mean near-surface temperatures aggregated at the country level with population density weights. In order to obtain these aggregation weights, we used country shapes (Burke et al., 2015) [2] in combination with spatial population density data from the History Database of the Global Environment (HYDE) version 3.1 (Klein Goldewijk et al., 2010) for the historical period and population scenario data consistent with the different SSP scenarios for the projection period (Jones and O'Neill, 2016b). The originally quinquennial population densities were linearly interpolated to annual values and conservatively upscaled (Jones, 1999) to the 0.5° CRU grid. These population space-time series were then masked with suitably rasterized country shapes and rescaled such that the resulting time-dependent country-wise 0.5° population density weights $w_{cij}$ satified $\sum_i w_{cij} = n_c$ for every country $c$ and year $j$, with $n_c$ being the number of grid cells that country $c$ occupies on the 0.5° CRU grid. Country-level temperature time series $\hat{T}_{cj}^{mp}$ were then obtained according to $\hat{T}_{cj}^{mp} = \frac{1}{n_c} \sum_i w_{cij} \tilde{T}_{ij}^{mp}$.

### A2. Calculation of damage costs

Here we summarize the description of the future extrapolation of the observed impact of changes in country-specific annual temperature on economic growth rates given in the supplement of Burke et al., 2015. Per-capita GDP in country $i$ in year $t$ emerges from the per-capita GDP of the previous year, the growth rate in absence of climate change $\eta_{it}$, which we take from the respective Shared Socioeconomic Pathways (SSPs), and the temperature impact on growth $\alpha_{it}$.

$$GDPcap_{it} = GDPcap_{it-1} \cdot (1 + \eta_{it} + \alpha_{it}) \tag{S4}$$

Note that this relation together with equation S5 is the core regression model in Burke et al., 2015. It assumes that climate damages have an impact on growth rates (rather than only on the level of GDP in a respective year) and thus have a persistent effect on future GDP levels of a

---

[2] Only for Indonesia and East Timor we used shapes from a different source since the independence of the latter from the former in 2002 was omitted in Burke et al.(Burke et al., 2015).

country. Several climate impacts can harm physical capital stocks and have long-lasting impacts on human capital and labor productivity, which causes an additional and more persistent impact on the rates of economic growth going beyond a purely instantaneous reduction of economic output. Even small changes in growth rates can result in significantly higher damages due to accumulation effects over time (Fankhauser and S.J. Tol, 2005; Moore and Diaz, 2015; Moyer et al., 2014). The ways in which climate change interacts with economic productivities, capital or labor stocks are complex and not yet fully understood (Huber et al., 2014); however, there is growing empirical evidence that increasing temperatures affect growth rates and not just output levels (Burke et al., 2015; Dell et al., 2012; Felbermayr and Gröschl, 2014; Hsiang, 2010).

The annual climate-induced growth deviation $\delta_{it}$ is determined by the empirical response function $h(T_{it})$. Since the response function is derived in differential terms, we need to subtract a reference value $h(\bar{T}_i)$ that corresponds to the initial year of the analysis.

$$\alpha_{it} = h(T_{it}) - h(\bar{T}_i) \tag{S5}$$

Using 2015 as the initial year, we start with $\bar{T}_i$ as the mean temperature for the period 2006–2024 of the no-further-warming scenarios (Section 1.2). We distinguish between rich and poor countries by choosing the respective empirical response function and regression parameters from the "base" case in Burke et al., 2015.

$$
\begin{aligned}
&h(T) \\
&= \begin{cases} \beta_1 T + \beta_2 T^2 & for\ rich\ countries, i.e.\ GDPcap_{it-1} > y^* \\ (\beta_1 + \beta_3)T + (\beta_2 + \beta_4)T^2 & for\ poor\ countries, i.e.\ GDPcap_{it-1} \le y^* \end{cases}
\end{aligned} \tag{S6}
$$

The separating value $y^*$ is the median per-capita GDP in 2010, i.e. at the end of the historical period in Burke et al., 2015. We do not assume that countries remain in the rich or poor category, but their response function is evaluated on the basis of each year's per capita GDP. Poor countries increasingly transition across the per-capita GDP threshold into the regime where the 'rich' specification for the response function is applied. The regression parameters are:

$$\beta_1 = 0.0089, \quad \beta_2 = -0.0003, \quad \beta_3 = 0.0165, \quad \beta_4 = -0.0005. \tag{S7}$$

$$se(\beta_1) = 0.0044, \quad se(\beta_2) = 0.0002, \tag{S8}$$

$$se(\beta_1 + \beta_3) = 0.0177, \quad se(\beta_2 + \beta_4) = 0.0004. \tag{S9}$$

The initial per-capita GDP values for 2015 ($GDPcap_{i,2015}$) are taken from the respective Shared Socioeconomic Pathways (SSPs). Note that in contrast Burke et al., 2015, (in the online source code) start their calculation in 2010 and initialize 2010 per-capita GDP values with country-specific average GDP values for the period 1980–2010. Hereby 2010 GDP values are assumed to be smaller than observed 2010 values or corresponding 2010 SSP values. Also the initial distribution of GDP values across countries is different; in particular some warm countries such as India, which have shown significant growth during 1980–2010, have a lower GDP share in global GDP when averaging the past decades. Hence, their relatively high GDP damages (due to high temperatures) receives more weight in our calculation which increases our total damage estimates by a factor of about two compared to Burke et al., 2015,for the RCP8.5 scenario. This

has been discussed and agreed upon with Marshall Burke at the side of a workshop in June 2016.

The estimated relationship between annual temperature fluctuations and the rate of change in GDP provided by Burke et al., 2015 implies that the GDP effect of a specific temperature deviation in one year would not be canceled out by the exactly opposite temperature deviation in the following year. Thus, assuming a stationary climate and translating its annual temperature fluctuations (as described by our no-further-warming scenario) into GDP deviations from a reference SSP scenario, not only leads to random fluctuations around the original SSP pathways but also to a systematic difference between the "perturbed SSP" pathway and the original one - a "pure fluctuation effect". Thus, to separate the pure effect of climate change, national SSP-based GDP trajectories are first perturbed by annual temperature fluctuation of the considered RCP. As the difference between these perturbed GDP time series and the original ones represents the climate change + fluctuation effect, we then subtract the fluctuation effect derived from the no-further-warming scenario runs to finally estimate the pure effect of climate change.

Cumulated damages in the 21$^{st}$ century depend on the timing of temperature increases, and not only on the temperature maximum, in particular because of the long-term nature of growth effects in the observed relation (Burke et al., 2015). Hence, all warming scenarios should build on sensible emissions and temperature pathways. However, the emissions of the RCP6.0 scenario develop in a peculiar way[3]: The historical trend is abruptly broken already in 2010 and emissions remain roughly constant from 2010 to 2030 before steeply increasing again until 2080 and then steeply decreasing after 2080. RCP6.0 emissions are actually below those of RCP4.5 and even below those of RCP2.6 scenario until after 2040 and 2020, respectively. The RCP6.0 emissions trajectory differs from those of the other RCPs and is not consistent with mitigation scenarios. Most mitigation scenarios show a smooth reduction of emissions. An early emissions' peaking or plateau combined with a later steep increase is rather unrealistic, as a transformation towards a low-carbon technology is usually not reversed in the second half of the century. We exclude the RCP6.0 scenario when calculating cumulated damages as a function of the global warming level and rely on the remaining three RCP scenarios and one emulated no-further-warming scenario.

Climate damages have been calculated for four climate scenarios, while REMIND mitigation scenarios have been estimated for 10 climate scenarios, which thus have a higher resolution in terms of global warming limits. Before we can combine GDP losses from damages and mitigation (on an annual and country level), the two sets of scenarios need to be harmonized. The relative GDP losses from damages for the RCPs are interpolated to the ten global warming limits of the mitigation cost scenarios such that mitigation and damage data refer to a consistent set of global warming limits. This interpolation is done for each year and each country using a linear regression. Fig. A1 shows results for SSP2, GCM IPSL-CM5A-LR and four major economies (China, India, Canada, Germany) that respond to global warming quite differently (losses and gains).

---

[3] See for example Fig. 2e in Meinshausen et al.(Meinshausen et al., 2011), which shows annual green-house-gas emissions for all four RCP scenarios.

The separately estimated annual country-specific GDP losses from detailed analyses of both climate impacts and climate mitigation can now be combined by reducing the reference GDP (without climate change) successively by the two relative GDP losses for each country and each year. The estimation of corresponding utility and the aggregation to a social welfare function is described in the main text of the manuscript.

### A3. Discussion of estimating future damages based on the historical relation

Burke et al., 2015 empirically derive a universal relation: for all countries the GDP response to annual country-specific temperature changes is described by the same function. This function is non-linear (concave) in the average temperature of a country, i.e. relatively cold countries showing GDP increases for warm years and already warm countries showing negative responses to warmer years. Economic productivity declines gradually with further warming, and this decline accelerates at higher temperatures. This non-linear function can be interpreted as a combination of linear responses to historical temperature changes for the different countries.

The crucial question is, whether the response function holds in projections of the long-term impact of global warming. If this is the case, the GDP response of a country with increasing average temperature would change according to the response function. If by contrast the relation changes under future climate change, the estimates of the optimal limit of global warming will likely change too. The extrapolation crucially builds on the stability of the relation. We present three arguments that support this assumption.

1) The observed relation seems to be quite robust over the historical period. Burke et al., 2015 show that the response function is fairly invariant in time, by dividing the data set and conducting two disjunct regression analyses for 1960–1989 and 1990–2010 (see Burke et al., 2015, Fig 2c).

2) Burke et al. conduct two disjunct regression analyses for poor and rich countries which show that the impacts on the growth rate are similar. To increase accuracy of our analysis, we apply the dedicated response functions for rich and poor countries, even though they do not differ by much. Note that impacts in the response function are calculated in relative terms (growth rate) and thus with increasing GDP in a country, the climate impacts increase in absolute terms.

When separating the data in rich and poor, the 50% confidence interval for the country response functions increase significantly, for poor countries especially at colder temperatures, for rich countries especially at warmer temperatures. This is due to the relatively small number of overall data points (N=6584) and because there is scarcer data for poor/rich-country at low/high temperatures. We consider these uncertainties when calculating optimal warming limits and extensively discuss this when presenting the results.

3) The annual country temperatures in the historical period (1960-2010) range from about -4°C to 29°C. Also with climate change, most countries would be in this temperature range. However, the assumption about the stability of the relationship loses validity with higher levels of global mean warming due to potential additional non-linear responses. While Burke et al., 2015 aim at comprehensively assessing economic

damages, some future climate impacts are likely to be missed or underestimated such that optimal warming limits would be even lower. The representation of climate damages as a simple function of annual temperatures and GDP neglects complex interactions between less aggregated economic damages, such as losses in specific economic sectors, and bio-physical impacts, such as floods or droughts, that will only unfold with further warming.

To understand the potential magnitude of this effect, we show Fig. A2 and A3. Therein we compare the temperature deviations for the historical period (dashed, deviations from 1960-2010 mean) that Burke et al., 2015 used for their regression and the future periods 2015-2050 and 2015-2100 (solid, deviations from the 20 year rolling mean around the reference year 2015) that we considered in the extrapolation of future climate damages. The distributions of annual mean temperature deviations are shown for five important countries (with cold, moderate and warm climate), for a "colder" GCM (GFDL-ESM2M, Fig. A2) and a "warmer" GCM (HadGEM2-ES, Fig. A3) and for RCP 2.6, RCP 4.5 and RCP 8.5.

For GFDL-ESM2M, there is significant overlap of historical and future temperature deviations for the RCP 2.6, RCP 4.5 and for the RCP 8.5 until 2050. Only the distribution of the long-term annual temperature changes for the RCP 8.5 (until 2100) is shifted to the warmer edge such that the overlap with the historical period is rather small. For the very warm GCM HadGEM2-ES, the long-term temperature changes of both the RCP 8.5 and already the RCP 4.5 are higher than for the historical period.

Historic temperature fluctuations are relatively large and in the same order of magnitude as the changes due to climate change until 2050 (rel. to 2015) even for the RCP 8.5. For pure rate of time preference >0 temperature changes during the first half of the century are more important than for the second half. In addition, it is the low end of warming where the assumption may be more justified and that is most relevant for our analysis, since we show a pronounced optimum around 2 degree.

## A4.    The REMIND model

This section describes the REMIND model. For more information, we refer to a paper and wiki that provides a detailed model description (Luderer et al., 2015, https://www.iamcdocumentation.eu/index.php/IAMC_wiki).

The REMIND model (Kriegler et al., 2017; Luderer et al., 2013b, 2015) represents the global energy-economy-climate system for 11 world regions and for the time horizon until 2100. REMIND represents five individual countries (China, India, Japan, United States of America, and Russia) and six aggregated regions formed by the remaining countries (European Union, Latin America, sub-Saharan Africa without South Africa, Middle East / North Africa / Central Asia, other Asia, Rest of the World). For each region, intertemporal welfare is optimized based on a Ramsey-type macro-economic growth model. The model explicitly represents trade in final goods, primary energy carriers, and in the case of climate policy, emission allowances and computes simultaneous and intertemporal market equilibria based on an iterative procedure. Macro-economic production factors are capital, labor, and final energy. REMIND uses economic output for investments in the macro-economic capital stock as well as consumption, trade, and energy system expenditures.

By coupling a macroeconomic equilibrium model with a technology-detailed energy model, REMIND combines the major strengths of bottom-up and top-down models. The macro-economic core and the energy system module are hard-linked via the final energy demand and costs incurred by the energy system. A production function with constant elasticity of substitution (nested CES production function) determines the final energy demand. For the baseline scenario, final energy demands pathways are calibrated to regressions of historic demand patterns. More than 50 technologies are available for the conversion of primary energy into secondary energy carriers as well as for the distribution of secondary energy carriers into final energy.

REMIND uses reduced-form emulators derived from the detailed land-use and agricultural model MAgPIE (Lotze-Campen et al., 2008; Popp et al., 2014) to represent land-use and agricultural emissions as well as bioenergy supply and other land-based mitigation options. Beyond CO2, REMIND also represents emissions and mitigation options of major non-CO2 greenhouse gases (EPA, 2013; Strefler et al., 2014).

Energy demand is an endogeneous variable to the model and determined as part of a macro-economic production function with constant elasticity of substitution (nested CES production function, see figure A5 which shows the structure and elasticities). This production function has been calibrated for consistency with historic trends, i.e. this specifically includes assumptions about future improvements of the productivity of input factors. For example, to calibrate baseline GDP, which is an endogenous result of the growth engine in REMIND, we adjust labor productivity parameters in an iterative procedure to e.g. reproduce the OECD's GDP reference scenarios. The REMIND scenarios (GDP, energy baseline demands) used for the manuscript at hand are calibrated such that they are close to a SSP2 scenario. The macro-economic core and the energy system module are hard-linked via the final energy demand and costs incurred by the energy system. Economic activity results in demand for final energy such as transport energy, electricity, and non-electric energy for stationary end uses. Final energy in the baseline scenarios (without climate change) for different sectors is calibrated to projections from the EDGE2 model (Energy Demand Generator, version 2) (e.g. Levesque et al., 2018).

REMIND computes the co-operative Pareto-optimal global equilibrium including inter-regional trade as the global social optimum using the Negishi method (Negishi, 1972), or the non-cooperative market solution among regions using the Nash concept (Leimbach et al., 2016). In the absence of non-internalized externalities between regions, these two solutions coincide.

## A5. Uncertainty in the mitigation cost estimates of energy-economy-climate models

Uncertainty in results of energy-economy-climate model is typically analysed by means of multi-model ensembles and in dedicated model-intercomparison projects, partly because structural differences matter. Note that Gillingham, Nordhaus et al., 2018 recently found that parametric uncertainty is more important than uncertainty in model structure for six models consisting of both cost-benefit and more detailed IAMs: DICE, FUND, GCAM, MERGE, IGSM, and WITCH). There are about one dozen well-established models in the global community of detailed integrated-assessment models, which shape the transformation pathway chapters as well as the summary for policy makers sections of the IPCC reports (Allen et al., 2018; IPCC, 2014a).

Figure A6 shows a comparison of REMIND mitigation costs used in this study with recent scenario results used and reported in the IPCC special report on 1.5°C. We calculated mitigation cost curves as a function of maximal global warming (until 2100). The costs are aggregated consumption losses relative to a baseline scenario (undiscounted for 2020-2050, left, and 2020-2100, right). The scenarios are filtered such that delayed action scenarios and constrained technology portfolio scenarios are removed. With decreasing warming limits, the models show steeply increasing costs that mark a threshold of further limiting global warming. Due to the high climate damages in Burke et al., 2015, this threshold determines to a large extent the optimal warming level in our study. Apart from the MERGE model, all models show this threshold in between 1.5 and 2°C. The REMIND model gives the median results of the five models reported in the IPCC SR1.5 Scenario Database (Huppmann et al., 2018).

Kriegler et al., 2015b conducted a diagnostic model study comparing several indicators including mitigation cost indicators on which we focus here. They show five indicators of mitigation costs (Figure 2, 3, 8, 10, 14 in Kriegler et al., 2015b). All cost indicators show significant differences across models. Results of the REMIND model are close to the across-model median. To conclude, using a different IAM model is likely to impact the result of our study, while the REMIND model is a somewhat representative model giving a middle of the road estimate for mitigation costs.

## A6. Assumption on the persistence of climate-change GDP impacts

Assumptions about the persistence of damages influence the total damage costs and thus the result of our study. Burke et al., 2015 (see Extended data figure 2a below) describe three cases with increasing damage without finding substantial empirical evidence for or against one of them: i) level effects, ii) one-year-growth effects (leading to persistent level effects), and iii) persistent growth effects. The latter case (as defined) includes additional future growth effects in response to an initial climate-change-related event. Hsiang and Jina, 2014, in a working paper, show such additional annual growth rate reductions until about 15-20 years after a disaster based on cyclone data. Assuming one-year-growth effects neglects both a potential recovery (towards level effects) and potential additional growth losses in following years (towards persistent growth effects).

Note that in addition a more-complex combined case is possible and seems plausible at least for e.g. severe extreme weather events: additional growth decreases could first increase the cumulated damage compared to one-year-growth effects, before recovery allows returning towards an original growth trajectory, which could be regarded as a larger multi-year level effect.

The assumption of one-year-growth effects leads to higher damages than pure level effects and to lower damages compared to persistent-growth effects. The relation to a larger multi-year level effect depends on its duration and amplitude (and also on the pure rate of time preference).

Trying to consider the full range of possible assumptions of persistence for deriving optimal warming levels is both valuable and challenging, as we argue below. This would ideally be based on a regression analysis that consistently derives damage estimates for different assumptions of persistence. One way is to increasingly include time lags into the regression analysis (next paragraph). As an alternative, one could assume a functional form of damage decay (e.g. exponential) and scan through different parameter values. Cumulated damages for a range of persistence values could then be combined with mitigation costs.

The magnitude and temporal structure of damages is uncertain, specifically when increasing the time horizon to derive insights on the persistence of damages. Burke et al., 2015 tried quantifying the latter by including lags in the regression. The resulting regression parameters get increasingly uncertain with more lags and the resulting damage impact is unclear. While for zero lags (=fully persistent damages) Burke's functional assumption of a parabolic (non-linear) response function can be confirmed very well within the uncertainty ranges (95% confidence interval) (see Burke et al., 2015, extended data figure below 2c, top right panel), this changes with the introduction of lags. Specifically for three- and five-year lags (2c, bottom), the median regression results are embedded in broad confidence intervals that allow for all sorts of functional response shapes and magnitudes. The median realization is below the x axis such that if a linear decline of dY/dT is assumed, the corresponding quadratic response function does not have a maximum anymore and also very cold countries would lose from any warming. The limited size of the data set (<10000) lets the signal get very weak when introducing additional variables (such as lags). While the cumulated damages (median values) reduce with introducing one lag pointing towards level effects, the impacts significantly increase with more lags. Given this and the increased uncertainty, we regard the lag-analysis (and question of persistence) as inconclusive.

Newell et al., 2018, in a recent working paper evaluate the performance of growth and level effect models with respect to the statistical significance of their results. They conclude that while the best-performing models are those that relate temperature to GDP levels, it cannot be precluded that growth-effect models are superior. Burke and Tanutama, 2019, in a recent working paper provide additional evidence for growth effects in a sub-national impact study. As mentioned above, Hsiang and Jina, 2014, in a working paper, even show additional annual growth rate reductions until about 15-20 years after a disaster based on cyclone data.

To sum up, the question of level vs. growth effects is relevant and open. We argue that a consideration of a range of persistence assumptions would ideally be based on a consistent empirical analysis that varies a persistence parameter. This is beyond the scope of our analysis. Here we assume one-year-growth effects and honestly communicate this important assumption. The effect of this assumption, in terms of how results are impacted cannot yet be answered in terms of magnitude and sign, as there is literature arguing towards both either level effects (which would reduce cumulated damages) or more persistent growth effects (increasing cumulated damages). We regard our focus and contribution as the combination of the default parameterization of damage response in Burke et al., 2015 (with high regression parameter robustness based on the assumption of one-year-growth effects) with global climate mitigation costs.

## Appendix Figures

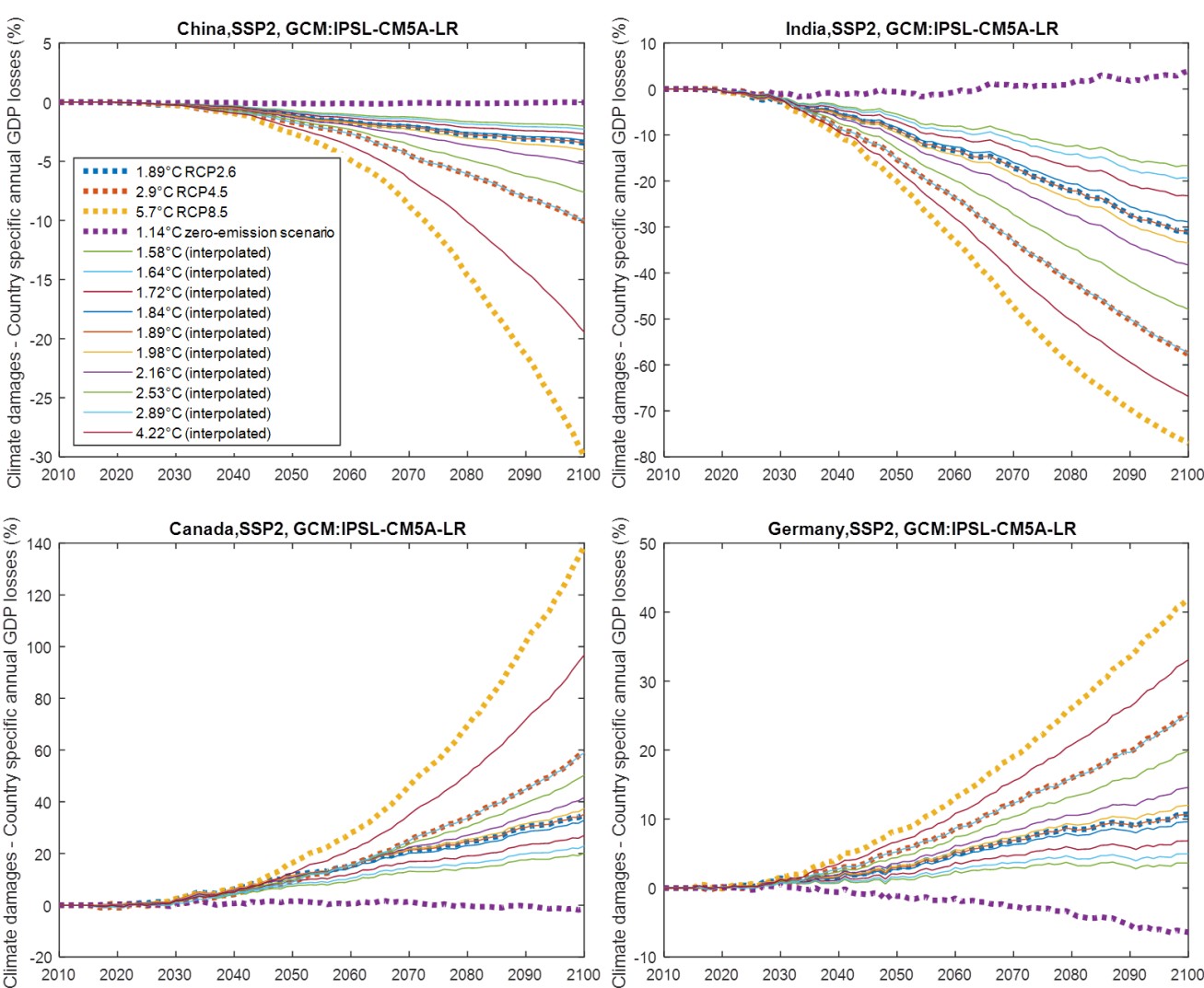

**Fig. A1: Annual GDP losses from climate change damages for four countries (China, India, Canada, Germany) and different maximum global warming levels (see legend) of three RCPs,**

the zero-emission scenario (dashed lines) and for 10 interpolated scenarios that correspond to the warming limits of the REMIND mitigation scenarios. Negative values correspond to losses.

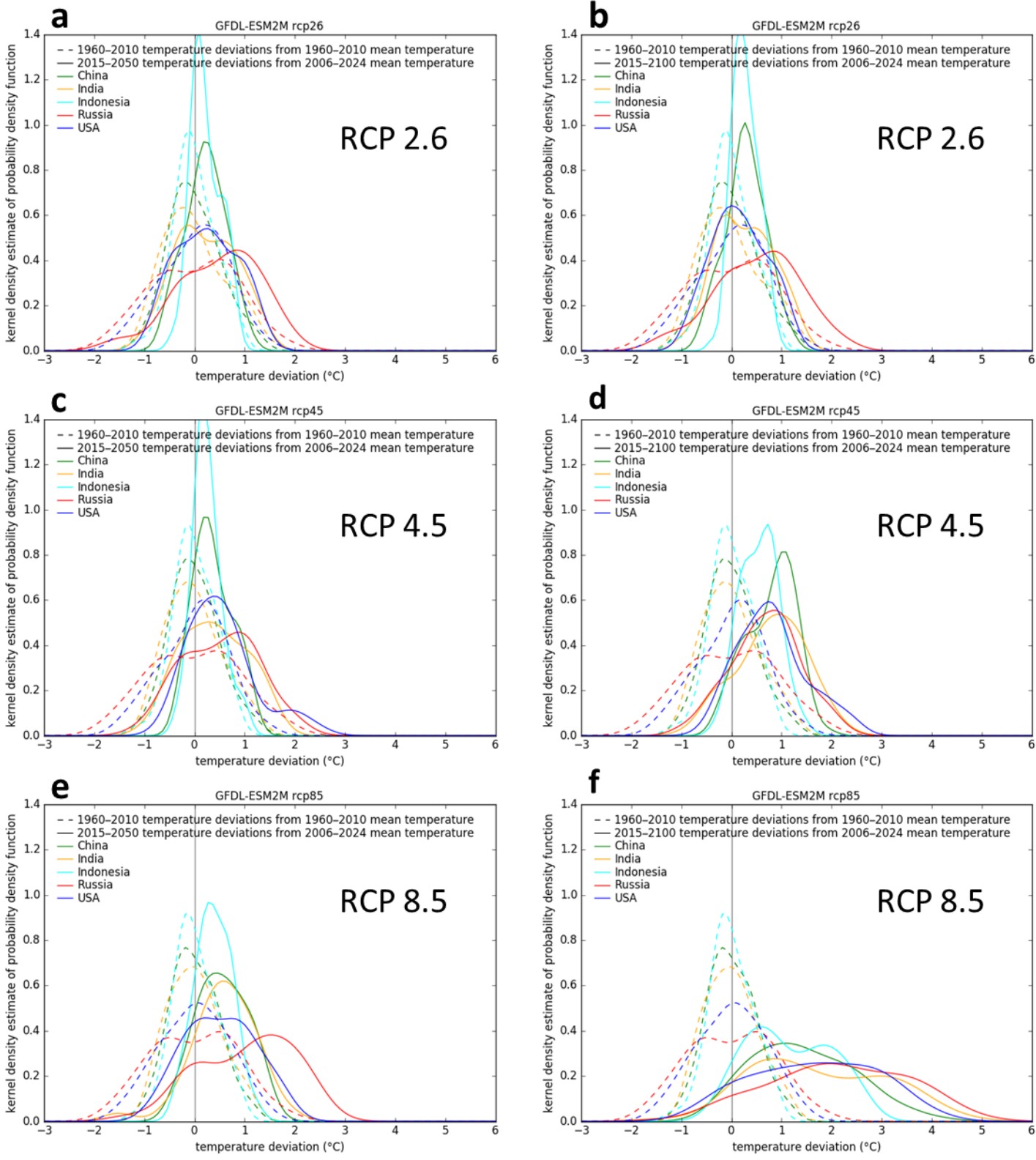

**Fig. A2: Comparing temperature deviations in the past and future.** From the GCM GFDL-ESM2M, a comparison of annual temperature deviations for the historical period (dashed) and future periods (solid) 2015-2050 (a, c, e) and 2015-2100 (b, d, f) for China, India, Indonesia, Russia and USA (colors), and for RCP 2.6 (a, b), RCP 4.5 (c, d) and RCP 8.5 (e, f).

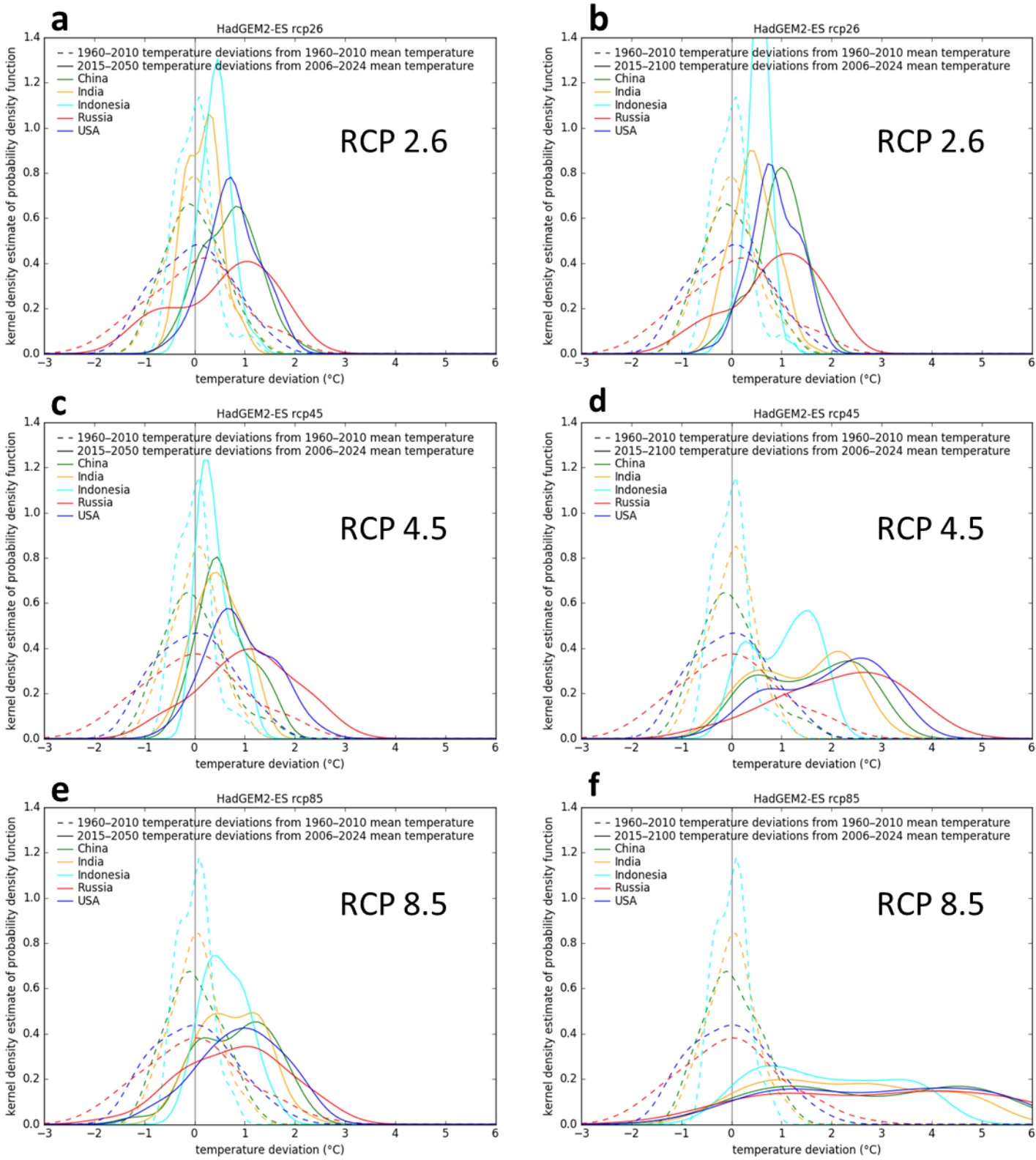

**Fig. A3: Same as Fig. A2, but from the "warmer" GCM HadGEM2-ES, a comparison of annual temperature deviations for the historical period (dashed) and future periods (solid) 2015-2050 (a, c, e) and 2015-2100 (b, d, f) for China, India, Indonesia, Russia and USA (colors), and for RCP 2.6 (a, b), RCP 4.5 (c, d) and RCP 8.5 (e, f).**

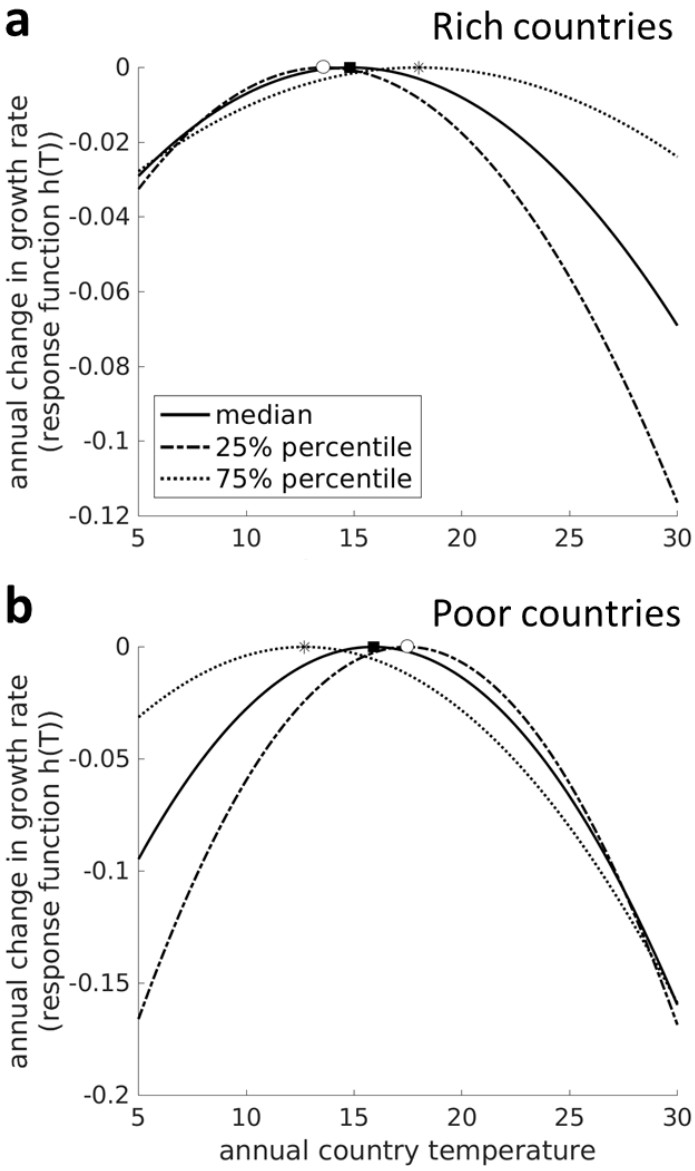

**Fig. A4: Effect of annual average country temperature on economic production (median, 25%, 75% percentile) for (a) rich and (b) poor countries based on regression parameters (median specification and standard errors) of the base case in Burke et al., 2015 (See also Appendix equations S7-S9).**

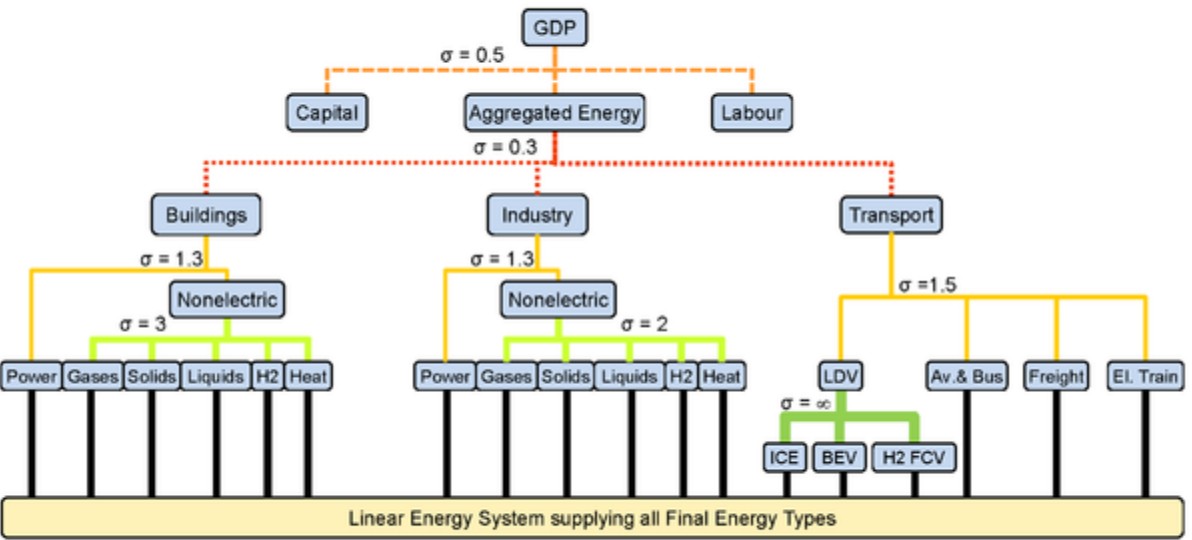

## ReMIND 1.7 CES Production Function

Abbreviations: *Heat: District heat & heat pumps, H2: Hydrogen, LDV: Light Duty Vehicle, ICE: Internal Combustion Engine, BEV: Battery Electric Vehicle, H2 FCV: Hydrogen Fuel Cell Vehicle, Av.& Bus: Aggregate of Aviation and Bus.*

**Fig. A5: CES production function of the REMIND 1.7 model.** Nested CES (*constant elasticity of substitution*) production functions determine the substitution of production factors, sector-specific energy demands, and energy carriers in the different levels of the function. The macro-economic production function is calibrated for consistency with historic trends, i.e. this specifically includes assumptions about future improvements of the productivity of input factors. Final energy in the baseline scenarios (without climate change) for different sectors is calibrated to projections from the EDGE2 model (Energy Demand Generator, version 2) (Levesque et al., 2018).

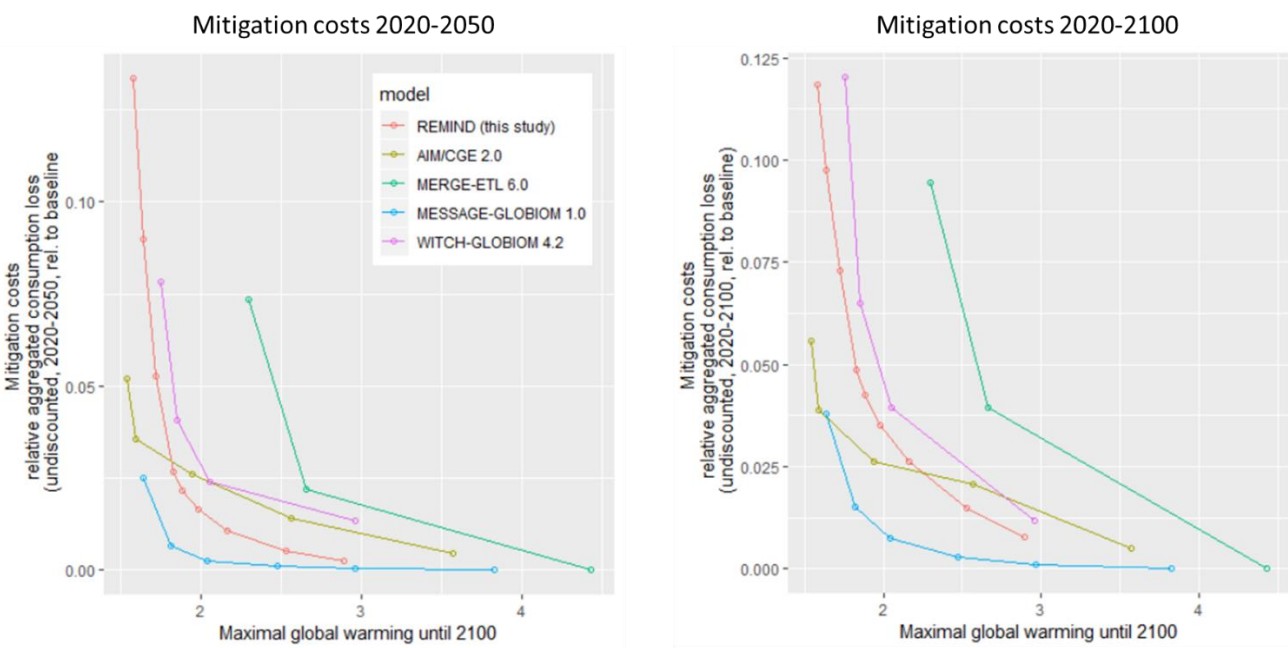

**Fig. A6: Comparison of REMIND mitigation costs used in this study with recent results of all models reported in the public dataset of the IPCC SR1.5 Scenario Database (Huppmann et al., 2018) used in the IPCC special report on 1.5°C. Mitigation cost curves are shown as a function of maximal global warming (until 2100). The costs are aggregated consumption losses relative to a baseline scenario (undiscounted for 2020-2050, left, and 2020-2100, right). The scenarios are filtered such that delayed action scenarios and constrained technology portfolio scenarios are removed.**