# Peer review of "The economically optimal warming limit of the planet"

_Earth System Dynamics, 2018_

## Referee Comment (RC1) · Moore (Referee) · 30 Jan 2019

This paper compares estimates of climate changes damages to GDP with costs of mitigation in order to identify the level of warming that minimizes the combined welfare losses. This is an important and innovative contribution. It is widely recognized that the cost-benefit IAMs used to make statements about optimal warming levels rely on outdated science. Much attention has been devoted recently to improving the representation of climate damages in these models. But the representation of mitigation costs is equally flawed. This paper makes a substantial advance in improving the representation of mitigation costs and combining with recent results of the magnitude of climate damages in order to assess optimal warming levels.

In addition to tackling an important question, the paper does a good job of examining sensitivity to preference parameters (specifically the pure rate of time preference and inequality aversion) and of communicating uncertainty from both climate models and damage function estimation. Nevertheless, I have five major concerns about the current manuscript – three related to the damage function and two to the mitigation cost estimates.

Damage Function

1. The paper uses results from the regression presented in Burke, Hsiang and Miguel (BHM). In the main specification, BHM assume that the growth rate effects estimated are permanent losses to growth. In fact, evidence for this is fairly weak – including one additional lag term substantially decreases the effect size and produces standard errors that overlap zero (BHM, Extended Data Figure 2c). More recent work has also shown suggested that the effects estimated by BHM are unlikely to be fully persistent (1). Given the indeterminacy in regression results regarding the question of whether these are growth or level effects, the authors should present using damage function specifications that include additional lag terms.

2. The authors are using a damage function specification that allows for different effects between rich and poor countries. As best I can determine, these classifications are fixed (i.e. a particular country remains on the poor damage function throughout the simulation, no matter how rich it gets). Instead the authors should account for the fact that poor countries are becoming richer over time, and therefore should eventually transition to the "rich" damage function after passing a certain income threshold.

3. I am unconvinced by the approach taken to try and "match" the climate damages based on RCP scenarios to the REMIND mitigation cost scenarios. The cumulative nature of the GDP impacts, combined with time discounting, means specific temperature trajectories could have a large effect on estimated damages. This means the interpolation approach taken here is almost certainly invalid since a least-cost approach to a peak warming of X degrees is different from any given RCP. Given the simplicity of damage estimates here (i.e a very simple function of population-weighted temperature change), I am not clear why the REMIND temperature trajectories themselves were not used to calculate damages, allowing for a direct mapping between damages and mitigation costs.

Mitigation Costs:

1. Given the main contribution of this paper is the comparison of mitigation costs with climate damages, far more information is needed on how mitigation costs are estimated in the REMIND model. At the moment there is only a very cursory paragraph describing this. Questions I would like answered include 1) how is energy demand estimated? 2) is the same SSP2 used to estimate energy demand? 3) how elastic is energy demand in the model and what is the empirical basis for this? 4) are these general equilibrium costs? 5) if not, to what extend are they likely to over- or under-estimate general equilibrium costs (particularly important for very ambitious mitigation targets)? 6) what are the uncertainties on the mitigation costs?

2. My main specific concern regarding mitigation costs surrounds the endogeneity of energy demand. A world where climate change damages have reduced GDP by up to 40% compared to baseline is a world with very different energy demand compared to a world without climate damages. The costs of reaching particular temperature targets would be correspondingly lower. A couple recent papers have examined this feedback between climate damages and the energy system (2, 3). My understanding is that this is not currently addressed. Ideally the authors would re-run REMIND, adjusting energy demand to account for the GDP damages.

Minor Comments

- The authors discuss extensively the limitations and uncertainties involved in the damage function calculation. I think it is important to have similar discussion about mitigation costs (e.g. why these might be either over or under estimates) and what effect that would have on the conclusions. Particularly given the fact that uncertainties in mitigation costs are not quantified.
- I don't think the term "non-economic" is appropriate for describing losses that are not captured by GDP. Economics captures any change that affects human welfare, including effects on ecosystems or health that would not be captured by GDP. The term "non-market" would be more appropriate.

References

1. Newell RG, Prest BC, Sexton SE (2018) *The GDP Temperature Relationship: Implications for Climate Change Damages* (Washington, D.C.).

2. Woodard DL, Davis SJ, Randerson JT (2019) Economic carbon cycle feedbacks may offset additional warming from natural feedbacks. *Proc Natl Acad Sci U S A* 116(3):759–764.

3. Bastien-Olvera BA (2019) Business-as-usual redefined: Energy systems under climate-damaged economies warrant review of nationally determined contributions. *Energy* 170:862–868.

---

## Referee Comment (RC2) · Anonymous Referee #2 · 6 Feb 2019

On the whole this is a useful paper that recognizes that climate change mitigation has costs and the optimal temperature target for how much we allow the world to warm depends on not just the damages of climate change but also the costs of mitigation. This is not a new insight but the paper represents a substantial advance over many of the IAMs that have been used in the past by incorporating a much more up-to-date approach to estimating the damage associated with a given temperature rise. On the mitigation side the paper includes a more robust estimation of how the costs of dealing with climate change will change with temperature and, it appears, a great deal more detail on how that will evolve with changes in technology.

I think that this paper usefully fills a gap in the existing literature by updating our understanding of the trade-off between mitigation costs and damages based on new science. However, there are several details of the implementation that concern me.

First, I share the concern pointed out by the first reviewer that the estimates from Burke et al. may not represent permanent declines in growth rates but rather represent a levels effect. Further, assuming these effects remain constant to the end of century, without accounting for adaptation, is problematic. Burke et al. suggest they see no evidence of changes in the relationship since the 1960s and take that as evidence of there being little adaptation but knowledge of climate change has changed a great deal since the last 40 years of the 20th century and it seems unlikely there will not be adaptation going forward. That also suggests a specification that does not treat damages as permanent declines in growth rates.

Second, I also would like more information about the REMIND model. It is a model of mitigation costs that is unfamiliar to me and the information included in the paper makes it difficult to assess whether it is appropriate.

Third, in the mitigation scenarios the authors assume "harmonized greenhouse gas emissions pricing as of 2020." That seems extremely unlikely to happen at this point. Similarly, while it is not clear when CCS is assumed to be widespread, any assumption that CCS is widespread in the very near future seems equally unrealistic. At the very least I'd like to see how the estimates change if these assumptions are relaxed and the time at which harmonized emissions pricing moves out into the future. I would expect that this increases the mitigation costs and so results in a higher temperature target but there may be offsetting effects (another reason to clarify the details of the REMIND model).

Fourth, I found the aggregation of costs and damages confusing. It seems that it would be better to estimate both costs and damages in the same framework rather than estimating them separately and then trying to aggregate. As the other reviewer points out, using the REMIND temperature pathways in the damage estimation seems the obvious way forward.

Fifth and finally, using a hard threshold to delineate between rich and poor seems fine but this should both be subject to robustness checks – what happens when the threshold is more or less strict? And countries should be allowed to move across the threshold overtime. As the author's point out, accounting for India's growth in the last few decades substantially increases the damage estimates in Burke et al. Despite climate change countries will continue to grow and as they grow richer they may become more adept at dealing with climate change. The estimates should allow for that.

Minor points:

- There were a number of typos throughout the manuscript. A quick read through should clean most of those up.
- I thought the figures 1 and 2 were clear and helpful but figure 3 may not have been necessary.

---

## Author Comment (AC1) · 4 Jun 2019

Reviewer 1

This paper compares estimates of climate changes damages to GDP with costs of mitigation in order to identify the level of warming that minimizes the combined welfare losses. This is an important and innovative contribution. It is widely recognized that the cost-benefit IAMs used to make statements about optimal warming levels rely on outdated science. Much attention has been devoted recently to improving the representation of climate damages in these models. But the representation of mitigation costs is equally flawed. This paper makes a substantial advance in improving the representation of mitigation costs and combining with recent results of the magnitude of climate damages in order to assess optimal warming levels.

In addition to tackling an important question, the paper does a good job of examining sensitivity to preference parameters (specifically the pure rate of time preference and inequality aversion) and of communicating uncertainty from both climate models and damage function estimation. Nevertheless, I have five major concerns about the current manuscript – three related to the damage function and two to the mitigation cost estimates.
Damage Function

*Thank you for your critical and excellent comments. We thoroughly revised the manuscript. We added significant discussion on assumptions and limitations of the study and included about 20 additional references. This includes a better explanation of the REMIND model and a discussion of uncertainty when modeling the mitigation cost side. For a point of comparison, we calculated aggregated mitigation cost curves for the latest scenario results of all models reported in the public dataset of the IPCC SR1.5 Scenario Database (https://data.ene.iiasa.ac.at/iamc-1.5c-explorer/) used in the IPCC special report on 1.5°C and show how the REMIND model results relate to it. Some of your concerns point to crucial questions of ongoing and future science that are best addressed in further studies, e.g. better understanding the persistence of damages and its effect on a cost-benefit analysis. In the revised document we now point to those directions for future research, while we more accurately convey the new aspects, insights and limitations/assumptions of our study.*

*With these responses we submit a manuscript that contains all tracked changes in response to the reviewer comments.*

1. The paper uses results from the regression presented in Burke, Hsiang and Miguel (BHM). In the main specification, BHM assume that the growth rate effects estimated are permanent losses to growth. In fact, evidence for this is fairly weak – including one additional lag term substantially decreases the effect size and produces standard errors that overlap zero (BHM, Extended Data Figure 2c). More recent work has also shown suggested that the effects estimated by BHM are unlikely to be fully persistent (1). Given the indeterminacy in regression results regarding the question of whether these are growth or level effects, the authors should present using damage function specifications that include additional lag terms.

*Thank you for pointing to this important assumption. Assumptions about the persistence of damages clearly influence the total damage costs and thus the result of our study. Burke et al. (see Extended data figure 2a below) differentiate three cases with increasing damage: i) level effects, ii) one-year-growth effects (leading to persistent level effects. You termed this 'full persistence'.), and iii) persistent growth effects. The latter case (as defined) includes additional future growth effects in response to an initial climate-change-related event. Hsiang and Jina, 2014, show such additional annual growth rate reductions until about 15-20 years after a disaster based on cyclone data. Assuming one-year-growth*

*effects not only neglects a potential recovery (towards level effects) but also excludes potential additional growth decreases in following years (towards persistent growth effects).*

*In addition, a more-complex combined case is possible and seems plausible at least for e.g. severe extreme weather events: additional growth decreases could first increase the cumulated damage compared to one-year-growth effects, before recovery allows returning towards an original growth trajectory, which could be regarded as a larger multi-year level effect.*

*The assumption of one-year-growth effects leads to higher damages than pure level effects and to lower damages compared to persistent-growth effects. The relation to a larger multi-year level effect depends on its duration and amplitude (and also on the pure rate of time preference).*

[Figure]

*Trying to consider the full range of possible assumptions of persistence is both valuable and challenging, as we argue below. In the manuscript, we now point to this as one possible next step for further research (which is of course already going on). This would ideally be based on a regression analysis that consistently derives damage estimates for different assumptions of persistence. One way is to increasingly include time lags into the regression analysis (next paragraph). As an alternative, one could assume a functional form of damage decay (e.g. exponential) and scan through different parameter values. Cumulated damages for a range of persistence values could then be combined with mitigation costs.*

*The magnitude and temporal structure of damages is uncertain, specifically when increasing the time horizon to derive insights on the persistence of damages. Burke et al. tried quantifying the latter by including lags in the regression. The resulting regression parameters get increasingly uncertain with more lags and the resulting damage impact is unclear. While for zero lags (=fully persistent damages) Burke's functional assumption of a parabolic (non-linear) response function can be confirmed very well within the uncertainty ranges (95% confidence interval) (see Burke's extended data figure below 2c, top right panel), this changes with the introduction of lags. Specifically for three- and five-year lags (2c, bottom), the median regression results (black) are embedded in broad confidence intervals that allow for all sorts of functional response shapes and magnitudes. The median realization is below the x axis such that if a linear decline of dY/dT is assumed, the corresponding quadratic response function does not have a maximum anymore and also very cold countries would lose from any warming. The limited size of the data set (<10000) lets the signal get very weak when introducing additional variables (such as lags). While the cumulated damages (median values) reduce with introducing one lag pointing towards level effects, the impacts significantly increase with more lags. Given this and the increased uncertainty, we regard the lag-analysis (and question of persistence) as inconclusive.*

[Figure]

*Newell et al. 2018 [2] (thank you for pointing us to this working paper) evaluate the performance of growth and level effect models with respect to the statistical significance of their results. They conclude that while the best-performing models are those that relate temperature to GDP levels, it cannot be precluded that growth-effect models are superior. Burke and Tanutama [3] (working paper) very recently provide additional evidence for growth effects in a sub-national impact study.*

*To sum up, the question of level vs. growth effects is relevant and open. We argue that a consideration of a range of persistence assumptions would ideally be based on a consistent empirical analysis that varies a persistence parameter. This is beyond the scope of our analysis. Here we assume one-year-growth effects and honestly communicate this important assumption. The effect of this assumption, in terms of how results are impacted cannot yet be answered in terms of magnitude and sign, as there is literature arguing towards both either level effects (which would reduce cumulated damages) or more persistent growth effects (increasing cumulated damages). We regard our focus and contribution as the combination of Burke's default parameterization of damage response (with high regression parameter robustness based on the assumption of one-year-growth effects) with global climate mitigation costs. We emphasize this now in the abstract, the introduction and the discussion and conclusion. Thanks again for inspiring us to deeper thinking and clarification on this issue. We also added this detailed discussion to the appendix.*

2. The authors are using a damage function specification that allows for different effects between rich and poor countries. As best I can determine, these classifications are fixed (i.e. a particular country remains on the poor damage function throughout the simulation, no matter how rich it gets). Instead the authors should account for the fact that poor countries are becoming richer over time, and therefore should eventually transition to the "rich" damage function after passing a certain income threshold.

*We have already been applying the suggested procedure. We had been stating that there is a fixed separating GDP value $y$ . Now we describe in more detail that the countries are newly evaluated against this separating value depending on each year's GDP level:*

*"The separating value $y$ is the median per-capita GDP in 2010, i.e. at the end of Burke's historical period. We do not assume that countries remain in the rich or poor category, but their response function is evaluated on the basis of each year's per capita GDP. Poor countries increasingly transition across the per-capita GDP threshold into the regime where the 'rich' specification for the response function is applied." (see supplement A2)*

3. I am unconvinced by the approach taken to try and "match" the climate damages based on RCP scenarios to the REMIND mitigation cost scenarios. The cumulative nature of the GDP impacts, combined with time discounting, means specific temperature trajectories could have a large effect on estimated damages. This means the interpolation approach taken here is almost certainly invalid since a least-cost approach to a peak warming of X degrees is different from any given RCP. Given the simplicity of damage estimates here (i.e a very simple function of population-weighted temperature change), I am not clear why the REMIND temperature trajectories themselves were not used to calculate damages, allowing for a direct mapping between damages and mitigation costs.

*We understand your concern. The reason why we cannot use the same temperature trajectories is that the REMIND model is not coupled with a GCM, but with MAGICC ('Model for the Assessment of Greenhouse Gas Induced Climate Change'). This is a reduced-complexity model that is often used by the IPCC, for key scientific publications, and by a number of Integrated Assessment Models. It calculates temperatures only with annual temporal resolution and for four earth sub-systems: ocean, land, northern and southern hemisphere*

*The resulting challenge is linking mitigation costs from the REMIND model and damage estimates based on a suite of GCM simulations. We have to find some kind of linking procedure because we rely on using GCMs for two reasons:*

1. *Using GCMs we can reflect climate modelling uncertainties via the variety of different approaches used and assumptions made by the different climate modelling teams.*
2. *Using GCMs allows to apply a high spatial resolution to then derive population-weighted temperatures based on spatially highly-resolved (0.5° x 0.5°) dynamic population projections and to then derive country-specific temperature data for each country to then derive country-specific GDP losses.*

*We agree with the reviewer that costs on both sides, mitigation and damages, depend on the temperature trajectories (i.e. on the timing of mitigation and damages). Ideally, we would have an identical set of temperature trajectories for a sufficient number of scenarios (~10) that span the full global warming range. However, running new GCM scenarios for several models is a tremendous effort that does not sufficiently strengthen the key point of this analysis: This study is showing that i) empirical estimates from a new research strand can be and need to be combined with detailed mitigation cost estimates (ideally from detailed integrated assessment models like REMIND), and that ii) a partial analysis combining empirical damages with one of the best state-of-the-art energy-economy-climate models shows that under assumptions on adaptation and persistence the "optimal" temperature increase is in the range of the Paris agreement for a wide range on normative assumptions on equality and time preference. Given that we could not build our analysis on an identical set of temperature trajectories for both mitigation and damage scenarios, we had to match both sides as described and made sure that their qualitative shape is similar. We had been excluding RCP60 from the analysis before as its emission trajectory is qualitatively different from the other RCPs and the REMIND scenarios.*

*The reviewer's concern points to a general weakness of the analysis, which is that it is a partial analysis. We acknowledge this weakness but nevertheless choose this approach for three reasons:*
1. *We want to make use of the high spatial resolution (i.e. countries across the world) that the damage response curve allows for. To our knowledge there is no model available that would allow for an integrated analysis with this spatial resolution.*
2. *An integrated analysis would be numerically very demanding. The partial analysis allows us to scan through multiple assumptions, i.e. normative assumptions on equality and time preference as well as accounting for climate uncertainties as given by the suite of GCMs.*
3. *The implementation of an empirical-based response function into an economic growth model (such as REMIND) is not straight forward as growth is endogenous to those models. It would require additional assumptions on the channel or pathway through which the damages are transmitted and a careful calibration to make sure the GDP effect is realized as shown in the empirical data. Moore and Diaz [4] demonstrate how this can be done for the DICE model using empirical damage estimates based on Dell et al.[5] and using two pathways: total factor productivity growth and capital depreciation empirical estimates.*

*There are weaknesses of a partial analysis. The biggest two we see and now discuss in the manuscript are:*
1. *Related to the above point of temperature trajectories: Only an integrated analysis would see trade-offs of mitigation and damage costs in time, i.e. when shifting mitigation efforts in time. The trajectory of mitigation efforts, temperature and damages changes while the maximum global warming level may remain unchanged. Unpublished studies with REMIND show that implementing damage (at the expense of losing empirical information) shifts mitigation efforts to earlier years to avoid short-term growth losses, in particular if those are persistent.*
2. *Only an integrated analysis can account for interactions of climate damages and climate mitigation, e.g. for energy supply (impacts of climate change on renewable resource potentials and variability) or impacts of climate change on energy demand*

*Both of these points cannot be considered in our partial analysis. We have now included a paragraph describing this limitation.*

*In our view, the scientific community should move towards trying to overcome these weaknesses, potentially by iteratively integrating mitigation and damage side more without losing key detail. One way of getting there is learning from a variety of (future) studies, more detailed partial ones, and less detailed integrated ones, and trying to synthesize robust findings to then derive a better understanding of the crucial aspects and detail that an integrated analysis should represent. We included >20 more references in the manuscript including a more extensive discussion of how our approach is different from other approaches taken so far.*

Mitigation Costs:
1. Given the main contribution of this paper is the comparison of mitigation costs with climate damages, far more information is needed on how mitigation costs are estimated in the REMIND model. At the moment there is only a very cursory paragraph describing this. Questions I would like answered include 1) how is energy demand estimated? 2) is the same SSP2 used to estimate energy demand? 3) how elastic is energy demand in the model and what is the empirical basis for this? 4) are these general equilibrium costs? 5) if not, to what extend are they likely to over-or under-estimate general equilibrium costs (particularly important for very ambitious mitigation targets)? 6) what are the uncertainties on the mitigation costs?

*We revised the manuscript and added more information on REMIND. Below we provide answers to your questions. For more details on model descriptions we refer to these sources:*
*[1] https://www.iamcdocumentation.eu/index.php/IAMC_wiki*
*[2] https://papers.ssrn.com/sol3/papers.cfm?abstract_id=2697070*

*The reason why we had put more emphasis on describing the damage side was that the empirical approach was rather new. The REMIND model, on the other hand, is one of about a dozen established state-of the-art energy-economy-climate models also referred to as IAMs. These IAMs are more detailed than cost-benefit IAMs such as the DICE model, i.e. they have more explicit process detail (e.g. ~50 energy conversion technologies). The detail of the REMIND model is mainly limited by future techno-economic parameter uncertainties and the numerical capacity of PIK's supercomputer. The REMIND model has been developed for ~15 years and is operated and further refined by a group of ~20 people at PIK. It is used in e.g. the IPCC reports (e.g. AR4, AR5, special report on renewables, special report on 1.5°C) and was recently chosen by the UN Finance Initiative to inform about climate-related transition risks (https://www.unepfi.org/wordpress/wp-content/uploads/2018/04/EXTENDING-OUR-HORIZONS.pdf).*

*Question 1) - 3)*

*Energy demand is an endogeneous variable to the model and determined as part of a macro-economic production function with constant elasticity of substitution (nested CES production function, see figure S5 which shows the structure and elasticities). This production function has been calibrated for consistency with historic trends, i.e. this specifically includes assumptions about future improvements of the productivity of input factors. For example, to calibrate baseline GDP, which is an endogenous result of the growth engine in REMIND, we adjust labor productivity parameters in an iterative procedure to e.g. reproduce the OECD's GDP reference scenarios. The macro-economic core and the energy system module are hard-linked via the final energy demand and costs incurred by the energy system. Economic activity results in demand for final energy such as transport energy, electricity, and non-electric energy for stationary end uses.*

[Figure]

**ReMIND 1.7 CES Production Function**

Abbreviations: *Heat: District heat & heat pumps, H2: Hydrogen, LDV: Light Duty Vehicle, ICE: Internal Combustion Engine, BEV: Battery Electric Vehicle, H2 FCV: Hydrogen Fuel Cell Vehicle, Av.& Bus: Aggregate of Aviation and Bus.*

*For the baseline scenario (without climate change), final energy in REMIND is calibrated to projections from the EDGE2 model (Energy Demand Generator, version 2). EDGE2 integrates econometric projections based on historical trends with scenario assumptions about long-term developments. The econometric projections play an important role in the short term while scenario assumptions rather influence the long-term behavior. The EDGE2 model covers six energy carriers— biomass, coal, electricity, liquids, gas, district heat —and six sectors —residential, commercial, industry, non-energy use, agriculture and fisheries, others. (https://www.iamcdocumentation.eu/index.php/Energy_demand_-_REMIND)*
*The REMIND scenarios (GDP, energy baseline demands) used for the manuscript at hand are calibrated such that they are close to a SSP2 scenario. For the buildings sector, a thorough explanation of how baseline buildings energy demand projections are derived and how they differ in the different SSP scenarios is given by*
*Levesque et al., How much energy will buildings consume in 2100? A global perspective within a scenario framework, Energy, Volume 148, 1 April 2018, Pages 514-527*
*https://www.sciencedirect.com/science/article/abs/pii/S0360544218301671*

*Question 4)*

*Yes. With REMIND, it is possible to compute the co-operative Pareto-optimal global equilibrium including inter-regional trade as the global social optimum using the Negishi method [6], or the non-cooperative market solution among regions using the Nash concept [7]. In the absence of non-internalized externalities between regions, these two solutions coincide.*

*Question 6)*

*Uncertainty in results of energy-economy-climate model is typically analyzed by means of multi-model ensembles and in dedicated model-intercomparison projects, partly because structural differences matter Note that in constrast Gillingham, Nordhaus et al. recently found that parametric uncertainty is more*

*important than uncertainty in model structure for six models consisting of both cost-benefit and more detailed IAMs: DICE, FUND, GCAM, MERGE, IGSM, and WITCH[8]. There are about one dozen well-established models in the global community of detailed integrated-assessment models, which shape the transformation pathway chapters as well as the summary for policy makers sections of the IPCC reports [9,10].*

Here we consider the cross-model variance in two ways. First, we now calculated aggregated mitigation cost curves for the latest model runs of all models used in the IPCC special report on 1.5°C to display the variance of results and how the REMIND model relates to it. We include a figure in the appendix of the paper. Second, we present results from *Kriegler et al.[11], who conducted a purely diagnostic model study comparing several indicators including mitigation cost indicators on which we focus here.*

*The below figure shows a comparison of REMIND mitigation costs used in this study with recent results of all models used in the IPCC special report on 1.5°C. We calculated mitigation cost curves as a function of maximal global warming (until 2100). The costs are aggregated consumption losses relative to a baseline scenario (undiscounted for 2020-2050, left, and 2020-2100, right). The scenarios are filtered such that delayed action scenarios and constrained technology portfolio scenarios are removed. With decreasing warming limits, the models show steeply increasing costs that mark* a threshold of further limiting global warming. Due to the high climate damages in Burke et al., this threshold determines to a large extend the optimal warming level in our study. Apart from the MERGE model, all models show this threshold in between 1.5 and 2°C. The *REMIND model gives the median results of the five models in the IPCC SR1.5 Scenario Database ([https://data.ene.iiasa.ac.at/iamc-1.5c-explorer/](https://data.ene.iiasa.ac.at/iamc-1.5c-explorer/)).*

*We included these figures and a discussion in the appendix of the paper.*

**Mitigation costs 2020-2050**

[Figure]

**Mitigation costs 20...**

[Figure]

*Kriegler et al.[11] conducted a purely diagnostic model study comparing several indicators including mitigation cost indicators on which we focus here. Below we show all five indicators of mitigation costs that are analyzed in the paper.*

*All cost indicators show significant differences across models. Results of the REMIND model are close to the across-model median. Using a different IAM model is likely to impact the result of our study. Expanding this study to using a different model, which would mean including other modeling teams to conduct the required model runs, is beyond our scope. The REMIND model seems to be a somewhat representative model giving a middle of the road estimate for mitigation costs.*

*We discuss the impact of this uncertainty on the results in the discussion part of the paper. We carefully conclude that expanding the analysis to a broader set of IAMs would change model-specific results by about +/- 0.2 °C for most models, while the median result remains roughly the same.*

[Figure]

**Fig. 10.** Development of the cost per abatement value (CAV) indicator over time and for two different tax scenarios: an exponentially increasing tax starting at $50 in 2010 (left panel) and a constant $200 carbon tax (right panel). Markers indicate the model for which the CAV was deduced. Solid lines indicate GE models, whereas dotted lines indicate PE models.

[Figure]

**Fig. 8.** Intertemporally aggregated mitigation costs (as a percentage of baseline consumption or GDP) for the periods 2010–2050 (left column) and 2010–2100 (right column). Cost measures differ for general (to the right of the vertical line) and partial equilibrium models (to the left). The scenarios listed from top to bottom in the figure legend are shown from left to right bars for each model.

[Figure]

**Fig. 2.** Relative abatement index over time and across models as deduced from the exponentially increasing carbon tax scenario starting at $50 in 2010 and increasing by 4% per year (left panel) and the constant $200 carbon tax scenario (right panel).

[Figure]

**Fig. 3.** Marginal abatement cost (MAC) curves for $CO_2$ fossil fuel and industry emissions as deduced from the exponentially increasing carbon tax scenarios in the diagnostic study. Emissions reductions as a fraction of baseline emissions are plotted against carbon price levels for the years 2045 (left panel) and 2080 (right panel). The emissions reductions in the carbon tax scenarios and the origin are combined into a MAC curve (dashed lines).

[Figure]

**Fig. 14.** Mitigation costs plotted against cumulated CO₂ FFI emissions reductions (fraction of baseline emissions) for 450 and 550 ppm CO₂eq climate stabilization scenarios (connected by dashed lines for a given model). Models are colored according to the classification in Table 3 (PE — low response: Red; PE — medium response: Dark Red; PE — high response: Green; GE — low response: Black; GE — high response: Yellow). To keep the y-axis at a scale that provides good visibility of most models' graphs, these panels exclude IMACLIM, which shows NPV policy costs of 6–7% for the 550 ppm scenario and 9–10% for the 450 ppm scenario for the periods 2010–2050 and 2010–2100.

2. My main specific concern regarding mitigation costs surrounds the endogeneity of energy demand. A world where climate change damages have reduced GDP by up to 40% compared to baseline is a world with very different energy demand compared to a world without climate damages. The costs of reaching particular temperature targets would be correspondingly lower. A couple recent papers have examined this feedback between climate damages and the energy system (2, 3). My understanding is that this is not currently addressed. Ideally the authors would re-run REMIND, adjusting energy demand to account for the GDP damages.

*Yes, the interaction of mitigation and damage costs is not part of the partial approach. This question is a very complex one that goes beyond the link through energy demand.*
*We included the following discussion in the main text of the paper.*

*We combine two partial analyses, for mitigation and damage costs. Not integrating them on the system level neglects three main interactions. First, climate induced reductions of economic productivity and associated reductions in energy demand would lead to reduced emissions without explicit mitigation measures [12,13], while climate impacts might reduce financial resources for climate mitigation. Second, climate impacts might change the future energy supply by their impact on renewable potentials and temporal variability (hydro, biomass, solar or wind power) and extreme weather events on energy infrastructure such as storm-induced transmission breakdowns and power outages or limited cooling water for nuclear or thermal plants (for further literature see this review: Cronin et al., 2018). Third, we did not analyze to what extent a full internalization of climate damages would shift the welfare optimal timing of mitigation to avoid short-term damages compared to a mitigation scenario that focuses only on limiting global warming. Reflecting those various interactions in an integrated study is complex and a future task to the scientific modeling communities. Accounting for these interactions requires a better process-understanding by which channels climate impact unfold and more empiric quantifications following pioneering work for individual processes e.g. energy demand [12,13]. Currently, the macro-level*

*temperature response identified by Burke et al., 2015 could not be broken down to individual processes. It even seems difficult and premature to conclude on the overarching magnitude or sign of climate impacts on the energy transitions and mitigation costs.*

Minor Comments
- The authors discuss extensively the limitations and uncertainties involved in the damage function calculation. I think it is important to have similar discussion about mitigation costs (e.g. why these might be either over or under estimates) and what effect that would have on the conclusions. Particularly given the fact that uncertainties in mitigation costs are not quantified.
Done in the discussion (and introduction) part of the paper.

- I don't think the term "non-economic" is appropriate for describing losses that are not captured by GDP. Economics captures any change that affects human welfare, including effects on ecosystems or health that would not be captured by GDP. The term "non-market" would be more appropriate.

Done. Thank you!

References
1. Newell RG, Prest BC, Sexton SE (2018) *The GDP Temperature Relationship: Implications for Climate Change Damages* (Washington, D.C.).
2. Woodard DL, Davis SJ, Randerson JT (2019) Economic carbon cycle feedbacks may offset additional warming from natural feedbacks. *Proc Natl Acad Sci U S A* 116(3):759–764.
3. Bastien-Olvera BA (2019) Business-as-usual redefined: Energy systems under climate-damaged economies warrant review of nationally determined contributions. *Energy* 170:862–868.

**REFERENCES in responses to both reviewers**

1. Hsiang, S. & Jina, A. *The Causal Effect of Environmental Catastrophe on Long-Run Economic Growth: Evidence From 6,700 Cyclones*. (National Bureau of Economic Research, 2014). doi:10.3386/w20352

2. Newell, R. G., Prest, B. C. & Sexton, S. E. The GDP-Temperature Relationship: Implications for Climate Change Damages. 63 (2018).

3. Burke, M. & Tanutama, V. *Climatic Constraints on Aggregate Economic Output*. (National Bureau of Economic Research, 2019). doi:10.3386/w25779

4. Moore, F. C. & Diaz, D. B. Temperature impacts on economic growth warrant stringent mitigation policy. *Nature Climate Change* **5**, 127–131 (2015).

5. Dell, M., Jones, B. F. & Olken, B. A. Temperature Shocks and Economic Growth: Evidence from the Last Half Century. *American Economic Journal: Macroeconomics* **4**, 66–95 (2012).

6. Negishi, T. *General equilibrium theory and international trade*. (North-Holland Publishing Company Amsterdam, London, 1972).

7. Leimbach, Schultes, Baumstark, Luderer & Giannousakis. Solution algorithms of large-scale Integrated Assessment models on climate change. *Annals of Operations Research* (2016).

8. Gillingham, K. *et al.* Modeling Uncertainty in Integrated Assessment of Climate Change: A Multimodel Comparison. *Journal of the Association of Environmental and Resource Economists* **5**, 791–826 (2018).

9. Allen, M. *et al. Summary for Policymakers. In: Global Warming of 1.5 °C an IPCC special report*. (IPCC, 2018).

10. IPCC. *Climate change 2014: mitigation of climate change : Working Group III contribution to the Fifth assessment report of the Intergovernmental Panel on Climate Change*. (Cambridge University Press, 2014).

11. Kriegler, E. *et al.* Diagnostic indicators for integrated assessment models of climate policy. *Technological Forecasting and Social Change* **90, Part A**, 45–61 (2015).

12. Bastien-Olvera, B. A. Business-as-usual redefined: Energy systems under climate-damaged economies warrant review of nationally determined contributions. *Energy* **170**, 862–868 (2019).

13. Woodard, D. L., Davis, S. J. & Randerson, J. T. Economic carbon cycle feedbacks may offset additional warming from natural feedbacks. *Proceedings of the National Academy of Sciences* **116**, 759–764 (2019).

14. Cronin, J., Anandarajah, G. & Dessens, O. Climate change impacts on the energy system: a review of trends and gaps. *Climatic Change* **151**, 79–93 (2018).

15. Burke, M., Hsiang, S. M. & Miguel, E. Global non-linear effect of temperature on economic production. *Nature* **527**, 235–239 (2015).

16. Rogelj, J. *et al.* Energy system transformations for limiting end-of-century warming to below 1.5 °C. *Nature Climate Change* **5**, 519–527 (2015).

17.     Strefler, J. *et al.* Between Scylla and Charybdis: Delayed mitigation narrows the space between

large-scale CDR and high costs. *Environmental Research Letters* (2018).

---

## Author Comment (AC2) · 4 Jun 2019

Reviewer 2

On the whole this is a useful paper that recognizes that climate change mitigation has costs and the optimal temperature target for how much we allow the world to warm depends on not just the damages of climate change but also the costs of mitigation. This is not a new insight but the paper represents a substantial advance over many of the IAMs that have been used in the past by incorporating a much more up-to-date approach to estimating the damage associated with a given temperature rise. On the mitigation side the paper includes a more robust estimation of how the costs of dealing with climate change will change with temperature and, it appears, a great deal more detail on how that will evolve with changes in technology.

*Thank you for your review and helpful comments! We carefully revised the manuscript and hope we could answer all your questions and concerns. With these responses we submit a manuscript that contains all tracked changes in response to the reviewer comments.*

I think that this paper usefully fills a gap in the existing literature by updating our understanding of the trade-off between mitigation costs and damages based on new science. However, there are several details of the implementation that concern me.

First, I share the concern pointed out by the first reviewer that the estimates from Burke et al. may not represent permanent declines in growth rates but rather represent a levels effect. Further, assuming these effects remain constant to the end of century, without accounting for adaptation, is problematic. Burke et al. suggest they see no evidence of changes in the relationship since the 1960s and take that as evidence of there being little adaptation but knowledge of climate change has changed a great deal since the last 40 years of the 20th century and it seems unlikely there will not be adaptation going forward. That also suggests a specification that does not treat damages as permanent declines in growth rates.

*Thank you for emphasizing this important assumption. We carefully thought through our assumptions and added significant discussion on the choice and implication for the results. We adjusted all crucial parts in the main text of the paper (abstract, introduction, method, discussion and conclusion) and added a section in the appendix A6.*

*Assumptions about the persistence of damages clearly influence the total damage costs and thus the result of our study. Burke et al. (see Extended data figure 2a below) differentiate three cases with increasing damage: i) level effects, ii) one-year-growth effects (leading to persistent level effects.), and iii) persistent growth effects. The latter case (as defined) includes additional future growth effects in response to an initial climate-change-related event. Hsiang and Jina, 2014, show such additional annual growth rate reductions until about 15-20 years after a disaster based on cyclone data. Assuming one-year-growth effects not only neglects a potential recovery (towards level effects) but also excludes potential additional growth decreases in following years (towards persistent growth effects).*

*In addition, a more-complex combined case is possible and seems plausible at least for e.g. severe extreme weather events: additional growth decreases could first increase the cumulated damage compared to one-year-growth effects, before recovery allows returning towards an original growth trajectory, which could be regarded as a larger multi-year level effect.*

*The assumption of one-year-growth effects leads to higher damages than pure level effects and to lower damages compared to persistent-growth effects. The relation to a larger multi-year level effect depends on its duration and amplitude (and also on the pure rate of time preference).*

[Figure]

*Trying to consider the full range of possible assumptions of persistence is both valuable and challenging, as we argue below. In the manuscript, we now point to this as one possible next step for further research (which is of course already going on). This would ideally be based on a regression analysis that consistently derives damage estimates for different assumptions of persistence. One way is to increasingly include time lags into the regression analysis (next paragraph). As an alternative, one could assume a functional form of damage decay (e.g. exponential) and scan through different parameter values. Cumulated damages for a range of persistence values could then be combined with mitigation costs.*

*The magnitude and temporal structure of damages is uncertain, specifically when increasing the time horizon to derive insights on the persistence of damages. Burke et al. tried quantifying the latter by including lags in the regression. The resulting regression parameters get increasingly uncertain with more lags and the resulting damage impact is unclear. While for zero lags (=fully persistent damages) Burke's functional assumption of a parabolic (non-linear) response function can be confirmed very well within the uncertainty ranges (95% confidence interval) (see Burke's extended data figure below 2c, top right panel), this changes with the introduction of lags. Specifically for three- and five-year lags (2c, bottom), the median regression results (black) are embedded in broad confidence intervals that allow for all sorts of functional response shapes and magnitudes. The median realization is below the x axis such that if a linear decline of dY/dT is assumed, the corresponding quadratic response function does not have a maximum anymore and also very cold countries would lose from any warming. The limited size of the*

*data set (<10000) lets the signal get very weak when introducing additional variables (such as lags). While the cumulated damages (median values) reduce with introducing one lag pointing towards level effects, the impacts significantly increase with more lags. Given this and the increased uncertainty, we regard the lag-analysis (and question of persistence) as inconclusive.*

[Figure]

*Newell et al. 2018 [2] evaluate the performance of growth and level effect models with respect to the statistical significance of their results. They conclude that while the best-performing models are those that relate temperature to GDP levels, it cannot be precluded that growth-effect models are superior. Burke and Tanutama[3] (working paper) very recently provide additional evidence for growth effects in a sub-national impact study.*

*To sum up, the question of level vs. growth effects is relevant and open. We argue that a consideration of a range of persistence assumptions would ideally be based on a consistent empirical analysis that varies a persistence parameter. This is beyond the scope of our analysis. Here we assume one-year-growth effects and honestly communicate this important assumption. The effect of this assumption, in terms of how results are impacted cannot yet be answered in terms of magnitude and sign, as there is literature arguing towards both either level effects (which would reduce cumulated damages) or more persistent growth effects (increasing cumulated damages). We regard our focus and contribution as the combination of Burke's default parameterization of damage response (with high regression parameter robustness based on the assumption of one-year-growth effects) with global climate mitigation costs. We emphasize this now in the abstract, the introduction and the discussion and conclusion. Thanks again for inspiring us to deeper thinking and clarification on this issue. We also added this detailed discussion to the appendix.*

Second, I also would like more information about the REMIND model. It is a model of mitigation costs that is unfamiliar to me and the information included in the paper makes it difficult to assess whether it is appropriate.

*Thank you! We added more paragraphs on the REMIND model in the main text and added two sections in the appendix: one on the REMIND model and one on mitigation cost uncertainties. Moreover we cite two main additional references for more detail:*
*[1] https://www.iamcdocumentation.eu/index.php/IAMC_wiki*
*[2] https://papers.ssrn.com/sol3/papers.cfm?abstract_id=2697070*

*In addition, we responded to each of the REMIND model questions that reviewer 1 was posing.*

Third, in the mitigation scenarios the authors assume "harmonized greenhouse gas emissions pricing as of 2020." That seems extremely unlikely to happen at this point. Similarly, while it is not clear when CCS is assumed to be widespread, any assumption that CCS is widespread in the very near future seems equally unrealistic. At the very least I'd like to see how the estimates change if these assumptions are relaxed and the time at which harmonized emissions pricing moves out into the future. I would expect that this increases the mitigation costs and so results in a higher temperature target but there may be offsetting effects (another reason to clarify the details of the REMIND model).

*A "harmonized greenhouse gas emissions pricing as of 2020" is the standard IAM modeling way to incentivize the global transition towards ambitious climate targets (in addition to other policy instruments such as subsidies). This does not imply that the modeling results are not valid if a global scheme does not exist in 2020 or later. Bertram et al., 2015, Nature Climate Change show that also an imperfect policy mix with delayed (until 2030) and fragmented carbon pricing can initiate a similar transformation at comparable mitigation costs.*

*At the same time, we frame our analysis by commenting on this also in the conclusion part: „a lack of political or societal will, partial interest groups and lobbying power, weak institutions, or insufficient international cooperation could hamper or delay a transition such that mitigation costs increase. Our analysis is meant to inform the ongoing international climate negotiations under the assumption that these barriers can be overcome."*

*For all technological aspects of the system transformation in the REMIND model and similar IAMs we refer to two main references that focus on very ambitious 1.5-2°C warming scenarios[9,16]. The key dependency of mitigation cost (and achievability of warming limits) are assumptions on the availability of CDR (carbon dioxide removal) including BECCS (biomass with CCS). The below figure derived with the REMIND model[17] shows the dependence of long term mitigation costs (=cumulated discounted consumption losses 2030-2100) on i) CDR availability (y-axis) and ii) short-term costs (upper x-axis: 2030 carbon pricing). This shows how reaching two temperature targets (left: likely 2°C, right: mean 1.5°C) depends on the availability of CDR and its tradeoff with short-term costs. The feasibility frontier is indicated by the red areas (very high mitigation cost). White spaces show areas beyond achievability. Short-term action (incentivized by high CO2 prices until 2030) can keep targets in reach even if less CDR is achievable in the long term.*

[Figure]

Fourth, I found the aggregation of costs and damages confusing. It seems that it would be better to estimate both costs and damages in the same framework rather than estimating them separately and then trying to aggregate. As the other reviewer points out, using the REMIND temperature pathways in the damage estimation seems the obvious way forward.

*We understand your concern. The reason why we cannot use the same temperature trajectories is that the REMIND model is not coupled with a GCM, but with MAGICC ('Model for the Assessment of Greenhouse Gas Induced Climate Change'). This is a reduced-complexity model that is often used by the IPCC, for key scientific publications, and by a number of Integrated Assessment Models. It calculates temperatures only with annual temporal resolution and for four earth sub-systems: ocean, land, northern and southern hemisphere.*

*The resulting challenge is linking mitigation costs from the REMIND model and damage estimates based on a suite of GCM simulations. We have to find some kind of linking procedure because we rely on using GCMs for two reasons:*

1. *Using GCMs we can reflect climate modelling uncertainties via the variety of different approaches used and assumptions made by the different climate modelling teams.*
2. *Using GCMs allows applying a high spatial resolution to then derive population-weighted temperatures based on spatially highly-resolved (0.5° x 0.5°) dynamic population projections and to then derive country-specific temperature data for each country to then derive country-specific GDP losses.*

*We agree with the reviewer that costs on both sides, mitigation and damages, depend on the temperature trajectories (i.e. on the timing of mitigation and damages). Ideally, we would have an identical set of temperature trajectories for a sufficient number of scenarios (~10) that span the full global warming range. However, running new GCM scenarios for several models is a tremendous effort that does not sufficiently strengthen the key point of this analysis: This study is showing that i) empirical estimates from a new research strand can be and need to be combined with detailed mitigation cost estimates (ideally from detailed integrated assessment models like REMIND), and that ii) a partial*

*analysis combining empirical damages with one of the best state-of-the-art energy-economy-climate models shows that under assumptions on adaptation and persistence the "optimal" temperature increase is in the range of the Paris agreement for a wide range on normative assumptions on equality and time preference. Given that we could not build our analysis on an identical set of temperature trajectories for both mitigation and damage scenarios, we had to match both sides as described and made sure that their qualitative shape is similar. We had been excluding RCP60 from the analysis before as its emission trajectory is qualitatively different from the other RCPs and the REMIND scenarios.*

Fifth and finally, using a hard threshold to delineate between rich and poor seems fine but this should both be subject to robustness checks – what happens when the threshold is more or less strict? And countries should be allowed to move across the threshold overtime. As the author's point out, accounting for India's growth in the last few decades substantially increases the damage estimates in Burke et al. Despite climate change countries will continue to grow and as they grow richer they may become more adept at dealing with climate change. The estimates should allow for that.

*We agree and have been applying the suggested procedure. We had been stating that there is a fixed separating GDP value $y$ . Now we describe in more detail that the countries are newly evaluated against this separating value depending on each year's GDP level, i.e. also countries in transition are moving into a regime, which uses the rich-country specification of the empirical response function.*

*"The separating value $y$ is the median per-capita GDP in 2010, i.e. at the end of Burke's historical period. We do not assume that countries remain in the rich or poor category, but their response function is evaluated on the basis of each year's per capita GDP. Poor countries increasingly transition across the per-capita GDP threshold into the regime where the 'rich' specification for the response function is applied." (see supplement A2)*

Minor points:
- There were a number of typos throughout the manuscript. A quick read through should clean most of those up.

*Thank you! All authors had a fresh read of the paper when revising it and we hope we covered this.*

- I thought the figures 1 and 2 were clear and helpful but figure 3 may not have been necessary.

*We agree and moved it to the appendix. Thank you!*

**REFERENCES in responses to both reviewers**

1. Hsiang, S. & Jina, A. *The Causal Effect of Environmental Catastrophe on Long-Run Economic Growth: Evidence From 6,700 Cyclones*. (National Bureau of Economic Research, 2014). doi:10.3386/w20352

2. Newell, R. G., Prest, B. C. & Sexton, S. E. The GDP-Temperature Relationship: Implications for Climate Change Damages. 63 (2018).

3.  Burke, M. & Tanutama, V. *Climatic Constraints on Aggregate Economic Output*. (National Bureau of Economic Research, 2019). doi:10.3386/w25779

4.  Moore, F. C. & Diaz, D. B. Temperature impacts on economic growth warrant stringent mitigation policy. *Nature Climate Change* **5**, 127–131 (2015).

5.  Dell, M., Jones, B. F. & Olken, B. A. Temperature Shocks and Economic Growth: Evidence from the Last Half Century. *American Economic Journal: Macroeconomics* **4**, 66–95 (2012).

6.  Negishi, T. *General equilibrium theory and international trade*. (North-Holland Publishing Company Amsterdam, London, 1972).

7.  Leimbach, Schultes, Baumstark, Luderer & Giannousakis. Solution algorithms of large-scale Integrated Assessment models on climate change. *Annals of Operations Research* (2016).

8.  Gillingham, K. *et al.* Modeling Uncertainty in Integrated Assessment of Climate Change: A Multimodel Comparison. *Journal of the Association of Environmental and Resource Economists* **5**, 791–826 (2018).

9.  Allen, M. *et al. Summary for Policymakers. In: Global Warming of 1.5 °C an IPCC special report*. (IPCC, 2018).

10.     IPCC. *Climate change 2014: mitigation of climate change : Working Group III contribution to the Fifth assessment report of the Intergovernmental Panel on Climate Change*. (Cambridge University Press, 2014).

11.     Kriegler, E. *et al.* Diagnostic indicators for integrated assessment models of climate policy. *Technological Forecasting and Social Change* **90, Part A**, 45–61 (2015).

12.     Bastien-Olvera, B. A. Business-as-usual redefined: Energy systems under climate-damaged economies warrant review of nationally determined contributions. *Energy* **170**, 862–868 (2019).

13.     Woodard, D. L., Davis, S. J. & Randerson, J. T. Economic carbon cycle feedbacks may offset additional warming from natural feedbacks. *Proceedings of the National Academy of Sciences* **116**, 759–764 (2019).

14.     Cronin, J., Anandarajah, G. & Dessens, O. Climate change impacts on the energy system: a review

of trends and gaps. *Climatic Change* **151**, 79–93 (2018).

15.     Burke, M., Hsiang, S. M. & Miguel, E. Global non-linear effect of temperature on economic

production. *Nature* **527**, 235–239 (2015).

16.     Rogelj, J. *et al.* Energy system transformations for limiting end-of-century warming to below 1.5

°C. *Nature Climate Change* **5**, 519–527 (2015).

17.     Strefler, J. *et al.* Between Scylla and Charybdis: Delayed mitigation narrows the space between

large-scale CDR and high costs. *Environmental Research Letters* (2018).